# The molecular basis underlying T cell specificity towards citrullinated epitopes presented by HLA-DR4

Tiing Jen Loh [1,4], Jia Jia Lim [1,4], Claerwen M. Jones [1], Hien Thy Dao[1], Mai T. Tran[1], Daniel G. Baker[2], Nicole L. La Gruta [1], Hugh H. Reid [1] ✉ & Jamie Rossjohn [1,3] ✉

CD4[+] T cells recognising citrullinated self-epitopes presented by HLA-DRB1 bearing the shared susceptibility epitope (SE) are implicated in rheumatoid arthritis (RA). However, the underlying T cell receptor (TCR) determinants of epitope specificity towards distinct citrullinated peptide antigens, including vimentin-64cit$_{59\text{-}71}$ and α-enolase-15cit$_{10\text{-}22}$ remain unclear. Using HLA-DR4-tetramers, we examine the T cell repertoire in HLA-DR4 transgenic mice and observe biased *TRAV6* TCR gene usage across these two citrullinated epitopes which matches with TCR bias previously observed towards the fibrinogen β −74cit$_{69\text{-}81}$ epitope. Moreover, shared *TRAV26-1* gene usage is evident in four α-enolase-15cit$_{10\text{-}22}$ reactive T cells in three human samples. Crystal structures of mouse TRAV6[+] and human TRAV26-1[+] TCR-HLA-DR4 complexes presenting vimentin-64cit$_{59\text{-}71}$ and α-enolase-15cit$_{10\text{-}22}$, respectively, show three-way interactions between the TCR, SE, citrulline, and the basis for the biased selection of *TRAV* genes. Position 2 of the citrullinated epitope is a key determinant underpinning TCR specificity. Accordingly, we provide a molecular basis of TCR specificity towards citrullinated epitopes.

The genetic susceptibility to autoimmune diseases is strongly linked to the human leukocyte antigen (HLA) class II locus[1]. In rheumatoid arthritis (RA), HLA-DR molecules associated with disease susceptibility were originally characterized by a common amino acid sequence in the HLA-DRβ1 chain, termed the HLA shared epitope (SE) with the sequence of either QKRAA, QRRAA, or RRRAA at positions 70–74[2]. Residues at positions 11, 71 and 74 were subsequently determined to confer the majority of RA risk associated with *HLA-DRB1* alleles with the *HLA-DRB1*04:01* allele having the strongest genetic association with the development of RA (odds ratio of 4.44)[3]. These residues are located within the antigen-binding cleft of the HLA molecule and determine the nature of the amino acid that can occupy the P4 pocket, and hence contribute to determining the peptides that are bound and presented to CD4[+] T cells. Negatively charged and polar/neutral

residues are favoured to occupy P4 whereas positively charged residues are disfavoured due to charge repulsion[4].

The HLA SE not only predisposes individuals to RA but also to the development of anti-citrullinated-protein antibodies (ACPAs)[5–9]. This is indicative that the T cell response is associated with the recognition of citrullinated proteins. Citrullination is a posttranslational modification (PTM) whereby arginine is deiminated to citrulline by peptidyl arginine deiminases (PAD)[10,11] with PAD2 and PAD4 the major isoforms associated with RA[12,13]. We previously provided a structural basis for the association of rheumatoid arthritis with SE allomorphs and citrullination, demonstrating that the conversion of positively charged arginine into neutral citrulline permits binding of citrullinated peptides to SE-containing HLA-DR allomorphs[4,14,15]. Hence, citrullination leads to the development of neoantigens, as PTM of peptides can confer the novel

[1]Infection and Immunity Program and Department of Biochemistry and Molecular Biology, Biomedicine Discovery Institute, Monash University, Clayton, Australia. [2]Janssen Research & Development, LLC, Horsham, Philadelphia, PA, USA. [3]Institute of Infection and Immunity, Cardiff University School of Medicine, Heath Park, Cardiff, UK. [4]These authors contributed equally: Tiing Jen Loh, Jia Jia Lim. ✉e-mail: hugh.reid@monash.edu; jamie.rossjohn@monash.edu

ability for them to bind to SE containing HLA-DR allomorphs. That these neo-antigenic peptides are not present in the thymus during central tolerance enhances their potential to induce an autoimmune response in the periphery. In RA synovial fluid cells, there is an increase in citrullination of proteins that drives the production and maintenance of ACPA, a hallmark of RA being detected prior to and after disease onset in 70% of patients[5–9,16–18]. Due to the cross-reactive nature of ACPA in recognising multiple citrullinated antigens, it is challenging to pinpoint an antigen specific T cell response in RA patients. In RA patients, several proteins found abundantly in the joint synovium and synovial fluid are citrullinated, including vimentin[19], type II collagen[20], fibrinogen[21], α-enolase[22], cartilage intermediate layer protein (CILP)[23], and tenascin C[24,25], some of which have been shown to be targets of the immune response[23,26]. Moreover, citrullinated peptide reactive CD4$^+$ T cells are present in transgenic mice with citrullinated vimentin and fibrinogen but not with their native forms[27,28]. We have previously demonstrated that citrullination of vimentin also alters its susceptibility to protease activity, thereby generating a stable self-epitope presented to T cells[4]. Furthermore, in HLA-DR4$^+$ RA patients, there is an increase in T helper 1 (Th1) effector memory CD4$^+$ T ($T_{em}$) cells specific for citrullinated epitopes[23,26,29], further implicating citrullinated epitope-specific CD4$^+$ T cells in disease pathogenesis.

We recently provided the structural basis for CD4$^+$ T cell receptor recognition of HLA-DR4 molecules presenting citrullinated fibrinogen[30]. Whether a common mechanism is observed for how TCRs recognize distinct citrullinated epitopes remains unclear[1,31]. Anti-Sa and anti-MCV antibodies are directed against citrullinated vimentin[19,32] and anti-CEP-1 antibodies are reactive against the citrullinated α-enolase CEP-1 epitope. Citrullinated vimentin, α-enolase, and fibrinogen antibodies are present in the range of 40% – 80% in ACPA-positive RA patients[33–37] suggesting these PTM proteins are autoantigens in RA. Here, we focus on investigating the T cell repertoire of citrullinated vimentin (Vim-64cit$_{59-71}$) and α-enolase (α-eno-15cit$_{10-22}$) peptides as well as deciphering the molecular basis of citrullinated peptide HLA (pHLA) recognition by TCRs isolated from T cells in the reactive repertoire. We provide insight into the specificity determinants underlying TCR recognition of different citrullinated self-antigens.

## Results

### Biased *TRAV* gene usage in HLA-DR4 mice immunized with citrullinated epitopes

Using humanised HLA-DR4 (*DRA1*01:01/DRB1*04:01*) transgenic mice[28] we investigated the T cell response generated to the HLA-DR4-restricted citrullinated self-epitopes, Vim-64cit$_{59-71}$ and α-eno-15cit$_{10-22}$ (GVYATXSSAVRLR and EIFDSXGNPTVEV, respectively, where X = Cit and underlined residues represent core binding region from P1 to P9 pockets of HLA-DR4)[23]. The human and mouse peptides of Vim-64cit$_{59-71}$ only differ at P2 where the alanine residue is replaced by valine residue in the mouse sequence. This difference is unlikely to influence peptide binding to HLA-DR4 as P2 is a solvent exposed residue. However, although it is a conservative substitution, this difference may potential influence TCR recognition. There is no difference in human and mouse peptides of α-eno-15cit$_{10-22}$. HLA-DR4 transgenic mice were co-immunised with the human Vim-64cit$_{59-71}$ and α-eno-15cit$_{10-22}$ peptides and, at 8 days post-immunisation, HLA-DR4$^{Vim-64cit59-71}$ and HLA-DR4$^{α-eno-15cit10-22}$ specific CD4$^+$ T cells were identified from draining lymph nodes (dLN) by binding to the HLA-DR4$^{Vim-64cit59-71}$ or HLA-DR4$^{α-eno-15cit10-22}$ tetramers, respectively. In all immunized mice, there was a clear expansion of CD4$^+$ tetramer$^+$ T cells in response to peptide immunisation, particularly to α-eno-15cit$_{10-22}$, as evidenced by the increased number and frequency of CD62L$^{lo}$ epitope-specific effector memory T ($T_{EFF}$) cells, relative to control immunised mice immunized with CFA-PBS. (Fig. 1a, b, Supplementary Fig. 1a).

We next characterised αβTCR gene usage within the HLA-DR4$^{Vim-64cit59-71}$ and HLA-DR4$^{α-eno-15cit10-22}$ tetramer$^+$ populations after immunization, which revealed substantial clonal expansions in both immune repertoires (Fig. 1c). Notably, immune repertoires to both the HLA-DR4$^{Vim-64cit59-71}$ and HLA-DR4$^{α-eno-15cit10-22}$ epitopes showed a strong preference for usage of the *TRAV6* gene elements. We previously observed this TCR bias in the immune response to the Fibβ-74$_{69-81}$ epitope in HLA-DR4 transgenic mice relative to the background CD4$^+$ T cell repertoire[30]. In addition, in the HLA-DR4$^{Vim-64cit59-71}$ immune repertoire there was preferential usage of a non-germline-encoded aspartic acid residue within the CDR3α loop, while in the HLA-DR4$^{α-eno-15cit10-22}$ specific TCR repertoire there was preferential usage (13/17 clonotypes) of *TRAJ* gene elements encoding a di-glycine (GG) motif. Expanded clones within each repertoire displayed distinct *TRBV* gene usage, which also differed between the HLA-DR4$^{Vim-64cit59-71}$ and HLA-DR4$^{α-eno-15cit10-22}$ immune repertoires, and thus there was no obvious preference for *TRBV* gene usage nor presence of CDR3β motifs. The epitope specificities of both expanded clones in the HLA-DR4$^{Vim-64cit59-71}$ immune repertoire and three of the expanded clones in the HLA-DR4$^{α-eno-15cit10-22}$ immune repertoire were confirmed by TCR expression in cell lines and staining TCR transfectants separately with HLA-DR4$^{Vim-64cit59-71}$, HLA-DR4$^{α-eno-15cit10-22}$ or control HLA-DR4$^{Fibβ-74cit69-81}$ tetramers (Fig. 1d). The TCRs from these expanded clones were specific to each epitope and did not cross-react.

### Isolation of HLA-DR4$^{α-eno-15cit10-22}$- and HLA-DR4$^{Vim-64cit59-71}$-specific TCRs from an ACPA$^+$ RA donor

To determine whether HLA-DR4$^{Vim-64cit59-71}$ and HLA-DR4$^{α-eno-15cit10-22}$-reactive CD4$^+$ T cells could be identified in HLA-DR4$^+$ humans, PBMCs from an ACPA$^+$ RA donor 2 were magnetically enriched using either the HLA-DR4$^{Vim-64cit59-71}$ or HLA-DR4$^{α-eno-15cit10-22}$, or HLA-DR4 influenza hemagglutinin epitope (HLA-DR4$^{HA}$) tetramers, using an established protocol[38,39]. One HLA-DR4$^{Vim-64cit59-71}$ tetramer binding CD4$^+$ T cell and three HLA-DR4$^{α-eno-15cit10-22}$ tetramer binding CD4$^+$ T cells were identified from the peripheral blood of this individual (Fig. 2a–c, Supplementary Fig. 1b). TCR sequencing of these tetramer$^+$ cells yielded one paired TCR α- and β-chain sequence from each sample (Fig. 2c), with only one or neither chain identified for the other two HLA-DR4$^{α-eno-15cit10-22}$ tetramer binding CD4$^+$ T cells. The specificity of the two TCRs, for which paired sequences were obtained, was confirmed by staining TCR transfected cells separately with HLA-DR4$^{Vim-64cit59-71}$, HLA-DR4$^{α-eno-15cit10-22}$ and control HLA-DR4$^{Fibβ-74cit69-81}$ tetramers (Fig. 2b).

### TCR affinity measurements

We expressed and purified five TRAV6$^+$ TCRs from the immune mouse T cell repertoire (HLA-DR4$^{Vim-64cit59-71}$ restricted TCRs A03 and A07 and HLA-DR4$^{α-eno-15cit10-22}$ restricted TCRs E02, E04, and E17), (Fig. 1c), and the two TRAV26-1$^+$ TCRs isolated from human RA$^+$ PBMCs (HLA-DR4$^{Vim-64cit59-71}$ restricted TCR RA2-A03 and HLA-DR4$^{α-eno-15cit10-22}$ restricted TCR RA2.7) (Fig. 2c). We then determined steady-state binding affinities ($K_D$) of the TCRs for their respective pHLA via surface plasmon resonance (SPR). The A03 TCR revealed a moderately high affinity for HLA-DR4$^{Vim-64cit59-71}$ with a $K_D$ of 6.2 μM, whereas the A07 TCR and RA2-A03 had moderate ($K_D$ 31.9 μM) and weak ($K_D$ 104.1 μM) affinities, respectively (Fig. 3a).

We did not observe any binding from SPR for three immune mouse TCRs restricted to HLA-DR4$^{α-eno-15cit10-22}$, which may be due to a weak affinity of these TCRs for HLA-DR4$^{α-eno-15cit10-22}$ beyond the sensitivity limits of SPR, where potential avidity effects observed with the tetramer do not translate (Fig. 1d). While the RA2.7 TCR exhibited a relatively high affinity of 7.1 μM for HLA-DR4$^{α-eno-15cit10-22}$, the SPR sensorgram displayed a low response with respect to the level of HLA-DR4$^{α-eno-15cit10-22}$ coupled to the Biacore sensor chip, which is indicative of an unstable HLA-peptide complex (Fig. 3b, left panel). To improve the pHLA stability, we mutated the α-eno-15cit$_{10-22}$ peptide at position 9, where valine was substituted with a glycine residue (V20G) which, based on previous immunopeptidomic studies, is a preferred anchor

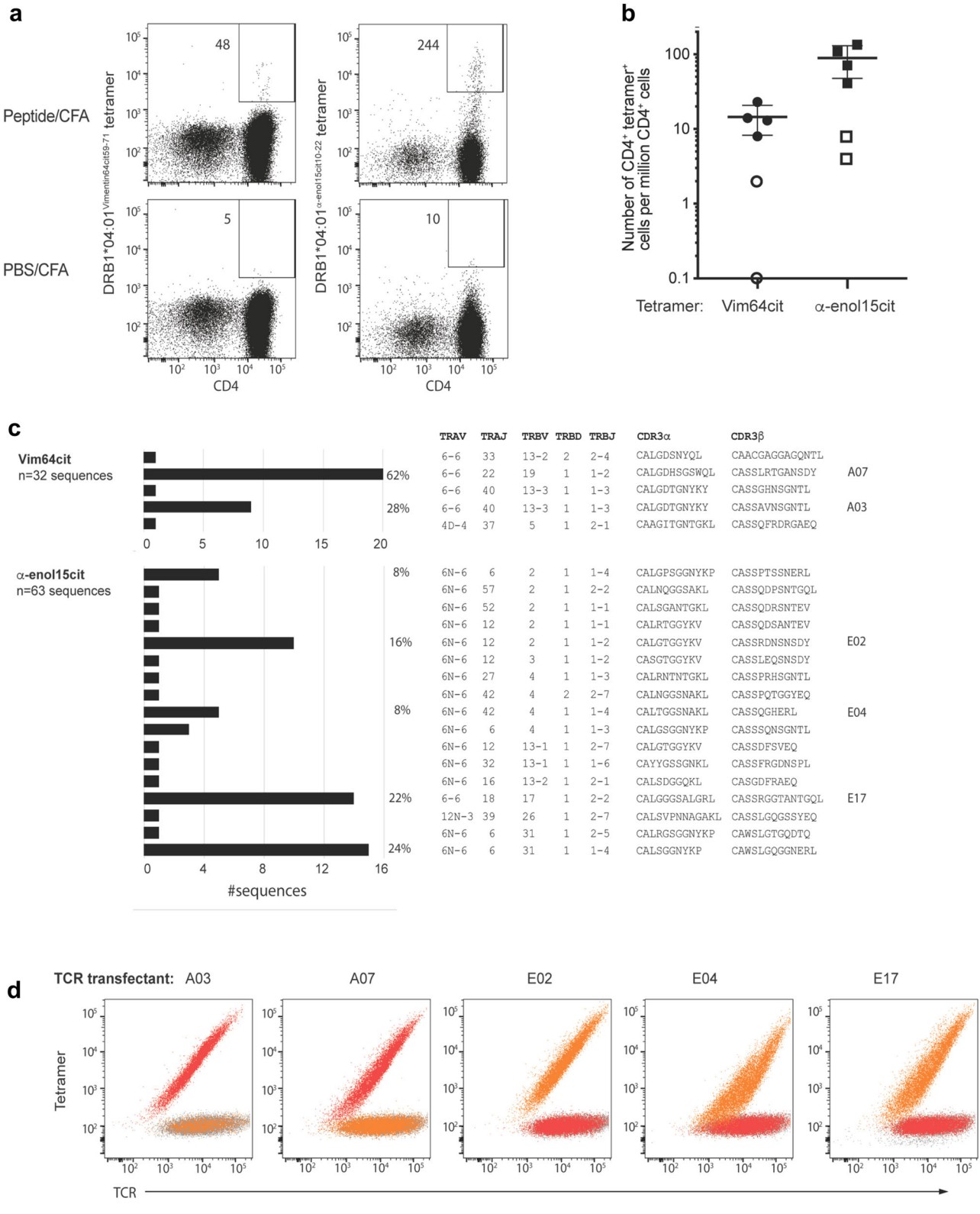

**Tetramer:** HLA-DRB1*04:01^Vimentin64cit59-71   HLA-DRB1*04:01^α-enolase15cit10-22   HLA-DRB1*04:01^Fibβ-74cit69-81

residue at P9 for HLA-DR4[4]. Subsequently, we generated an SKW-3 T cell line transduced with the RA2.7 TCR and determined upregulation of the early T cell activation marker CD69 and downregulation of CD3 as the readout for activation of RA2.7 TCR upon recognition of HLA-DR4^α-eno-15cit10-22 on BLCL antigen presenting cells fed the α-eno-15cit$_{10-22}$

peptide. T cell reactivity toward both wildtype and V20G mutant peptide was significant at a low peptide concentration of 0.08 μg/mL and reached maximal response at 10 μg/mL (Fig. 3c–f). Hence, the V20G mutant peptide had a similar effect on the activation of TCR compared to wildtype peptide albeit the response to the V20G mutant

**Fig. 1 | HLA-DR4$^{Vim-64cit59-71}$- and HLA-DR4$^{α-eno-15cit10-22}$-specific CD4$^+$ T cells in immune HLA-DR4 transgenic mice. a** Representative HLA-DR4$^{Vim-64cit59-71}$ and HLA-DR4$^{α-eno-15cit10-22}$ tetramer staining on CD4$^+$ T cells from the draining lymph nodes (dLN) of HLA-DR4 mice immunized 8d previously with a 50:50 mix of Vimentin64cit$_{59-71}$ and α-enolase-15cit$_{10-22}$ peptides (upper plots) or PBS (lower plots) emulsified in CFA. Gating strategies shown in Supplementary Fig. 1. Numbers in dot plots represent total number of CD4$^+$ tetramer$^+$ cells after magnetic enrichment and acquisition of entire sample. **b** Frequency of CD4$^+$ HLA-DR4$^{Vim-64cit59-71}$ (Vim-64cit)- or HLA-DR4$^{α-eno-15cit10-22}$ (α-eno-15cit)-tetramer positive CD4$^+$ cells in the dLN of mixed peptide- (closed symbols; $n = 4$) or PBS- (open symbols; $n = 2$) immunized HLA-DR4 mice. Symbols represent data from individual mice obtained

in two independent experiments. Horizontal bars indicate mean ± SD. **c** HLA-DR4$^{Vim-64cit59-71}$- and HLA-DR4$^{α-eno-15cit10-22}$-specific TCRαβ repertoires isolated from the dLN of a peptide-immunized HLA-DR4 mouse ($n = 1$ with 32 and 63 sequences, respectively), showing the frequency of individual clones and the TCR gene segment usage and CDR3 amino acid sequence for each clone. CDR3 nucleotide sequences are listed in Supplementary Table 10. TCRs marked A07, A03, E02, E04 and E17 were selected for further analysis. **d** 293 T cells transiently co-transfected with a HLA-DR4$^{Vim-64cit59-71}$ specific TCR (A03 and A07) or HLA-DR4$^{α-eno-15cit10-22}$ specific TCR (E02, E04, E17) and CD3γδεζ were stained with HLA-DR4$^{Vim-64cit59-71}$ tetramer (red), HLA-DR4$^{α-eno-15cit10-22}$ tetramer (orange) or control HLA-DR4$^{Fibβ-74cit69-81}$ tetramer (charcoal). Source data are provided as a Source Data file.

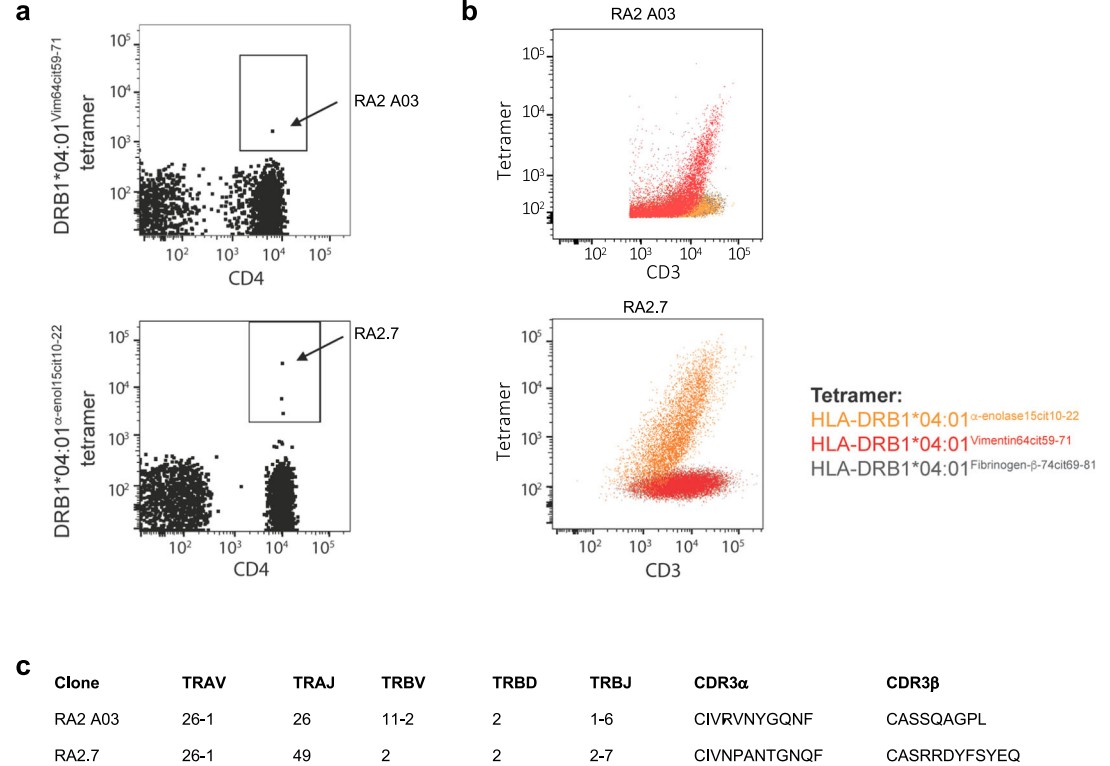

| Clone | TRAV | TRAJ | TRBV | TRBD | TRBJ | CDR3α | CDR3β |
|---|---|---|---|---|---|---|---|
| RA2 A03 | 26-1 | 26 | 11-2 | 2 | 1-6 | CIVRVNYGQNF | CASSQAGPL |
| RA2.7 | 26-1 | 49 | 2 | 2 | 2-7 | CIVNPANTGNQF | CASRRDYFSYEQ |

**Fig. 2 | HLA-DR4$^{Vim-64cit59-71}$- and HLA-DR4$^{α-eno-15cit10-22}$-specific CD4$^+$ T cells in RA donor. a** HLA-DR4$^{Vim-64cit59-71}$ and HLA-DR4$^{α-eno-15cit10-22}$ tetramer staining on CD4$^+$ T cells post tetramer-based magnetic enrichment of PBMC from a HLA-DR4$^+$ ACPA$^+$ RA donor. Dot plots show CD14⁻CD19⁻live CD3$^+$ CD4$^+$ cells. Gating strategy is shown in Supplementary Fig. 1b. Arrows indicate cells designated RA2 A03 and RA2.7. **b** 293 T cells transiently co-transfected with RA2.7 TCR and CD3γδεζ or SKW-3

T cells transiently co-transfected with RA2 A03 TCR and CD3γδεζ were stained with either HLA-DR4$^{α-eno-15cit10-22}$ tetramer (orange), HLA-DR4$^{Vim-64cit59-71}$ tetramer (red) or control HLA-DR4$^{Fibβ-74cit69-81}$ tetramer (charcoal). **c** TCR gene segment usage and CDR3 amino acid sequence for RA2 A03 and RA2.7 clones. CDR3 nucleotide sequences are listed in Supplementary Table 10.

was higher likely due to its effective concentration being higher as it binds with higher affinity, and hence more stably, to HLA-DR4. We then determined the affinity of RA2.7 TCR for HLA-DR4 presenting α-eno-15cit$_{10-22V20G}$ (HLA-DR4$^{α-eno-15cit10-22V20G}$) and obtained a $K_D$ of 7.1 µM with more than double the response units obtained for the native peptide, indicating that the V20G mutation conferred stability to the pHLA complex and with limited or no influence on TCR recognition (Fig. 3b). We also generated SKW-3 T cell line transduced with the A03 or A07 mouse-human hybrid TCRs and showed T cell reactivity toward Vim64cit 59-71 peptide at a low peptide concentration (Supplementary Fig. 2a, b).

The affinity of the mouse and human citrullinated epitope reactive TCRs determined here was well within the range generally observed of autoreactive TCRs (i.e. -$K_D$ > 30 µM) as well as those observed for TCRs reactive to exogenous antigen and PTM self-antigens (i.e. -$K_D$ 1–30 µM).

## Biased *TRAV26-1* gene usage in HLA-DR4$^{α-eno-15cit10-22V20G}$ tetramer binding TCRs isolated from RA donors and healthy controls

Using the modified HLA-DR4$^{α-eno-15cit10-22V20G}$ tetramer we successfully isolated and obtained paired TCRαβ sequence of a further 11 CD4$^+$ T cells from the peripheral blood of ACPA$^+$ RA donor 2 as well as two CD4$^+$ T cells from another RA donor 3 and 20 and 15 CD4$^+$ T cells, respectively, from two HLA-DR4$^+$ healthy donors (Supplementary Table 1). While the vast majority of TCR in the repertoire were unique, analysis of *TRAV* and *TRBV* gene element usage (Fig. 3g) highlighted preferential use of the *TRAV26-1* gene element in all four donors. Interestingly, this is the *TRAV* gene element used by both the HLA-DR4$^{α-eno-15cit10-22V20G}$-specific RA2.7 TCR and the HLA-DR4$^{Vim-64cit59-71}$-specific RA2-A03 TCR (Figs. 2c and 3b). All four donors also had several TCRs within their HLA-DR4$^{α-eno-15cit10-22V20G}$ tetramer-binding repertoire that used the *TRBV2* gene element (Fig. 3g), also used by the RA2.7 TCR. Selected TCR, whose epitope specificity was confirmed by

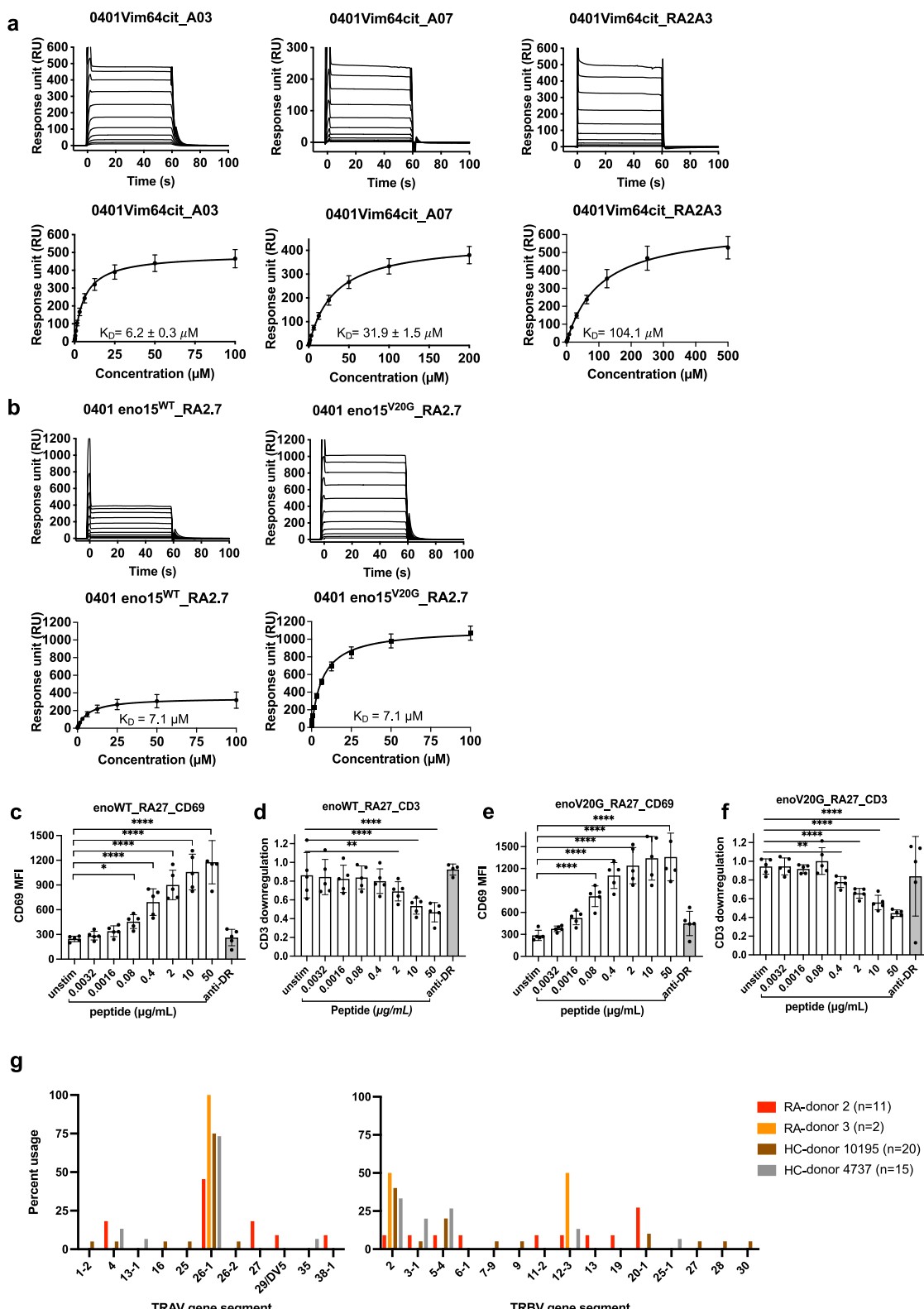

tetramer staining of TCR transfectants, affirmed usage of the *TRAV26-1* or *TRBV2* gene element for recognition of the HLA-DR4[α-eno-15cit10-22V20G] epitope (Supplementary Table 1).

## TCR recognition of HLA-DR4[Vim-64cit59-71]

To understand the molecular basis for the *TRAV6* gene preference from the immune mouse T cell repertoire, we first determined the complex structures of the A03 (TRAV6-6⁺-TRBV13-3⁺) and A07 (TRAV6-6⁺-TRBV19⁺) TCRs bound to HLA-DR4[Vim-64cit59-71] at 2.65 Å and 2.75 Å resolution, respectively (Fig. 4a, Supplementary Table 2). Both the A03 and A07 TCRs docked canonically, ~67° and 61° across the central region of the antigen binding cleft, respectively (Fig. 4b). The total buried surface area (BSA) for the A03 TCR was ~2010 Å² and A07 TCR was ~1665 Å², falling within the typical range of relative BSA values observed for TCR-pMHC II structures[40] (Supplementary Table 3). Both TCRs revealed a higher contribution to binding by Vα-chain,

**Fig. 3 | Affinity analysis of TCR-HLA-DR4$^{\text{Vim-64cit59-71}}$/α-eno-15cit10-22 interactions, antigen reactivity of RA2.7 TCR towards HLA-DR4$^{\alpha\text{-eno-15cit10-22}/\alpha\text{-eno-15cit10-22V20G}}$ and TCR gene segment usage of ACPA$^+$ RA donor and healthy control CD4$^+$ T cells binding to HLA-DR4$^{\alpha\text{-eno-15cit10-22V20G}}$ tetramer. a** Binding of TCRs A03, A07 and RA2A03 to HLA-DR4$^{\text{Vim-64cit59-71}}$ and (**b**) Binding of RA2.7 TCR to HLA-DR4$^{\alpha\text{-eno-15cit10-22}/\alpha\text{-eno-15cit10-22V20G}}$. For K$_D$ determination in (**a**) and (**b**) all data were derived from two or more independent experiments, with A03 TCR ($n = 7$), A07 TCR ($n = 6$), RA2A3 TCR ($n = 2$), RA2.7 ($n = 2$) and curve fits using a single ligand binding model. For each concentration the points represent the mean and error bars correspond to SD. Expression of CD69 on the surface of RA2.7 TCR transduced SKW-3 cell lines stimulated overnight with serial dilution of (**c**) α-eno-15cit 10-22 peptide-pulsed BLCL 9031 or (**e**) α-eno-15cit 10-22V20G peptide- pulsed BLCL 9031. Expression of CD3 on the surface of RA2.7 TCR transduced SKW-3 cell lines stimulated overnight with serial dilution of (**d**) α-

eno-15cit 10-22 peptide-pulsed BLCL 9031 or (**f**) α-eno-15cit 10-22V20G peptide- pulsed BLCL 9031. For (**c–f**) the black dots are presented as the mean fluorescence intensity (MFI) of average of duplicated values from three independent experiments. Anti-HLA-DR4 antibody used as control. For (**c–f**) *P*-values were determined by one-way ANOVA with Dunnett's multiple comparison testing, \**P* < 0.05, \*\**P* < 0.01, \*\*\**P* < 0.002, \*\*\*\**P* < 0.0001 and error bars represent ± s.e.m. **g** TCR *TRAV* and *TRBV* gene segment usage of CD4$^+$ T cells isolated from the peripheral blood of ACPA$^+$ RA donor 2, ACPA$^-$ RA donor 3, and HLA-DR4$^+$ healthy control (HC) donors 10195 and 4737 using HLA-DR4$^{\alpha\text{-eno-15cit10-22V20G}}$ tetramer. All TCR clones were unique except for one TRAV26-1$^+$ TRBV5-4$^+$ clone from HC donor 4727 for which two sequences were isolated (Supplementary Table 1). Source data are provided as a Source Data file.

comprising 54.5% and 63.7% of total BSA, respectively (Fig. 4b), likely explaining the *TRAV6* bias in the HLA-DR4$^{\text{Vim-64cit59-71}}$ CD4$^+$ T cell response.

The CDR1α, CDR2α, CDR3α, and framework (FW) region of the A03 TCR contributed 20.7%, 9.6%, 20.8% and 3.4% to the BSA, respectively, at the A03 TCR-HLA-DR4$^{\text{Vim-64cit59-71}}$ interface and, similarly, the A07 TCR contributed 23.1%, 8.7%, 29.6% and 2.3% to the BSA, respectively (Fig. 4b, Supplementary Table 3). The CDR1α and CDR3α loops made the most interactions with the peptide and HLA-DR4. The side chain of Tyr$^{36}$α from the CDR1α loop packed against Thr$^{77}$β of HLA-DR4 and formed extensive van der Waals (vdW) interactions (Fig. 4c, Supplementary Table 4). These interactions were further extended by hydrogen bonds from the neighbouring CDR1 Asn$^{38}$α interacting with HLA-DR4 Gln$^{70}$β and the CDR1 Ala$^{35}$α main chain contacting HLA-DR4 His$^{81}$β. CDR2α-HLA-DR4β interactions were apolar in character and A03 TCR FW Lys$^{55}$α formed a salt bridge with Asp$^{66}$β from HLA-DR4β chain (Fig. 4c, Supplementary Table 4). The CDR3α loop sat centrally atop HLA-DR4 and contacted both the α- and β- chains of HLA-DR4. Polar mediated interactions between CDR3 Thr$^{109}$α with HLA-DR4 Gln$^{70}$β and CDR3 Tyr$^{112}$α and Lys$^{113}$α with Glu$^{55}$α and Gln$^{57}$α of HLA-DR4 further supported the CDRα interactions (Fig. 4D, Supplementary Table 4). The brace-like conformation of CDR3α is supported by Asp$^{108}$α forming hydrogen bonds with Asn$^{111}$α (Fig. 4d, Supplementary Table 4). Despite distinct CDR3α sequences, the A07 TCR revealed a similar docking pattern of CDRα loops that contact both the HLA α- and β-chains (Fig. 4f, g) and maintained the brace-like conformation through vdW interactions between CDR3 Asp$^{108}$α and Trp$^{113}$α (Fig. 4g, Supplementary Table 5). The CDR3 His$^{109}$α wedged between the side chains of Gln$^{70}$β, Ala$^{73}$β and Thr$^{77}$β of the HLA β-chain helix and formed hydrogen bonds to Gln$^{70}$β and Thr$^{77}$β while CDR3 Ser$^{112}$α and Trp$^{113}$α contacted the HLA-DR4 α-chain (Fig. 4g, Supplementary Table 5). Accordingly, the interactions suggested a basis for the *TRAV6*+ bias, and conserved Asp residue in the CDR3α loop observed in the HLA-DR4$^{\text{Vim-64cit59-71}}$ restricted CD4$^+$ T cell response.

Within the A03 TCR-HLA-DR4$^{\text{Vim-64cit59-71}}$ complex, the TRBV13-3$^+$ β-chain contributed 45.5% to the BSA with the CDR1β, CDR2β and FWβ (6.3%, 10.7%, and 1%, respectively) of interacting with the HLA-DR4 α-chain while the CDR3β loop (27.5%) interacted with the HLA-DR4 β-chain (Fig. 4e, Supplementary Table 3). In contrast, for the TRBV19$^+$ A07 TCR, the β-chain contributed 36.3% to the BSA of the ternary complex, with contacts mediated solely by the CDR3β loop. Here, the CDR3β loop in the A07 TCR docked in the middle of the antigen binding groove, enabling simultaneous contact with the HLA-DR4 α- and β-chains (Fig. 4h, Supplementary Table 5), while leaving the adjacent CDR1β and 2β oriented away from antigen binding groove.

In the ternary complex, the A03 TCR contacted the side chains of solvent exposed residues (P2-Ala, P3-Thr, P4-Cit, P5-Ser, and P8-Ala) of the Vim-64cit$_{59\text{-}71}$ peptide, as well as forming interactions with the main chains of P1-Tyr, P3-Thr, P6-Ser and P8-Ala (Fig. 4i). The A07 TCR

made similar peptide interactions except that the non-germline encoded CDR3 loops did not interact with the P3 side chain and P6 main chain of the Vim-64cit$_{59\text{-}71}$ peptide (Fig. 4j). Overall, the HLA-DR4$^{\text{Vim-64cit59-71}}$ restricted TCRs exhibited close contact with the peptide as multiple main chain interactions were observed. Collectively, both the A03 and A07 TCRS exhibit highly conserved interactions with HLA-DR4 and the Vim-64cit$_{59\text{-}71}$ epitope, albeit utilizing distinct residues in CDR3α and CDR3β loops.

## Structural basis for biased *TRAV6$^+$* gene usage

*TRAV6$^+$* gene usage was been consistently enriched in three distinct and non-cross-reactive immune mouse TCR repertoires specific for HLA-DR4-bound citrullinated epitopes; namely fibrinogen β (Fibβ-74cit$_{69\text{-}81}$)[30], Vim-64cit$_{59\text{-}71}$ and α-eno-15cit$_{10\text{-}22}$ (Fig. 1c). To understand the basis of biased *TRAV6$^+$* gene usage in TCR binding to these different citrullinated epitopes presented by HLA-DR4, we compared the structures of the A03 TCR-HLA-DR4$^{\text{Vim-64cit59-71}}$ complex with the M134 TCR-HLA-DR4$^{\text{Fibβ-74cit69-81}}$ complex[30]. The TRAV6D-6$^+$ M134 TCR has nearly identical germline-encoded CDR1α and CDR2α to the TRAV6-6$^+$ A03 TCR except Thr$^{27}$α is replaced by alanine in the CDR1α loop (Fig. 5a), with this residue not being involved in any interactions with peptide or HLA-DR4 (Fig. 5b, c). In both TRAV6$^+$ TCR-pHLA crystal structures, the CDR1α is stabilized by CDR2α (residues 57-59) and CDR3α (residue 107-108) through Ala$^{35}$/Ile$^{30}$, Tyr$^{36}$α and Asn$^{38}$α with multiple hydrogen bonds and vdW forming a rigid unit. The TRAV6$^+$ CDR1α was focused on interactions with the P2 residue (Fig. 5b, c). In contrast to the Vim-64cit$_{59\text{-}71}$ complexes, in the TRAV6$^+$ M134 TCR-HLA-DR4$^{\text{Fibβ-74cit69-81}}$ complex the α-chain only contributed 26.5% to the interface BSA[30]. The long and positively charged P2 arginine in the Fibβ-74cit$_{69\text{-}81}$ complex pushed the TCR α-chain further away from the peptide binding groove resulting in reduced TCR α-chain contact with HLA-DR4 and peptide (Fig. 5b, c), thereby explaining the reduced contribution in BSA. Despite reduced HLA-DR4 interactions with distinct epitopes, the conserved contacts of HLA-DR4 Glu$^{55}$α, Gln$^{57}$α, Gly$^{58}$α, and Thr$^{77}$β from HLA-DR4 with TCR in both complexes indicated these residues are required for TRAV6$^+$ TCR recognition of HLA-DR4 and are likely to be involved in the biased gene usage observed in response to this HLA allomorph (Fig. 5b, c).

## Energetic basis of biased *TRAV6$^+$* gene usage

We used the A03 TCR-HLA-DR4$^{\text{Vim-64cit59-71}}$ complex to undertake an alanine-scanning mutagenesis and SPR approach to study the energetic basis for the *TRAV6$^+$* bias and *TRBV13-3$^+$* usage. We chose eleven point mutations of the A03 TCR for analysis based on TCR-pHLA interactions within the ternary structure. Notably, alanine substitution of conserved germline encoded residues within the A03 TCR α-chain (CDR1α Ala$^{35}$, Tyr$^{36}$, Asn$^{38}$, and CDR2α Ile$^{57}$), had deleterious effects on HLA-DR4$^{\text{Vim-64cit59-71}}$ recognition ($\geq$ 10-fold reduced affinity compared to wildtype TCR) (Fig. 5d, Supplementary Fig. 3). These four residues were involved in pHLA interaction as well as inter-CDRα contacts, as

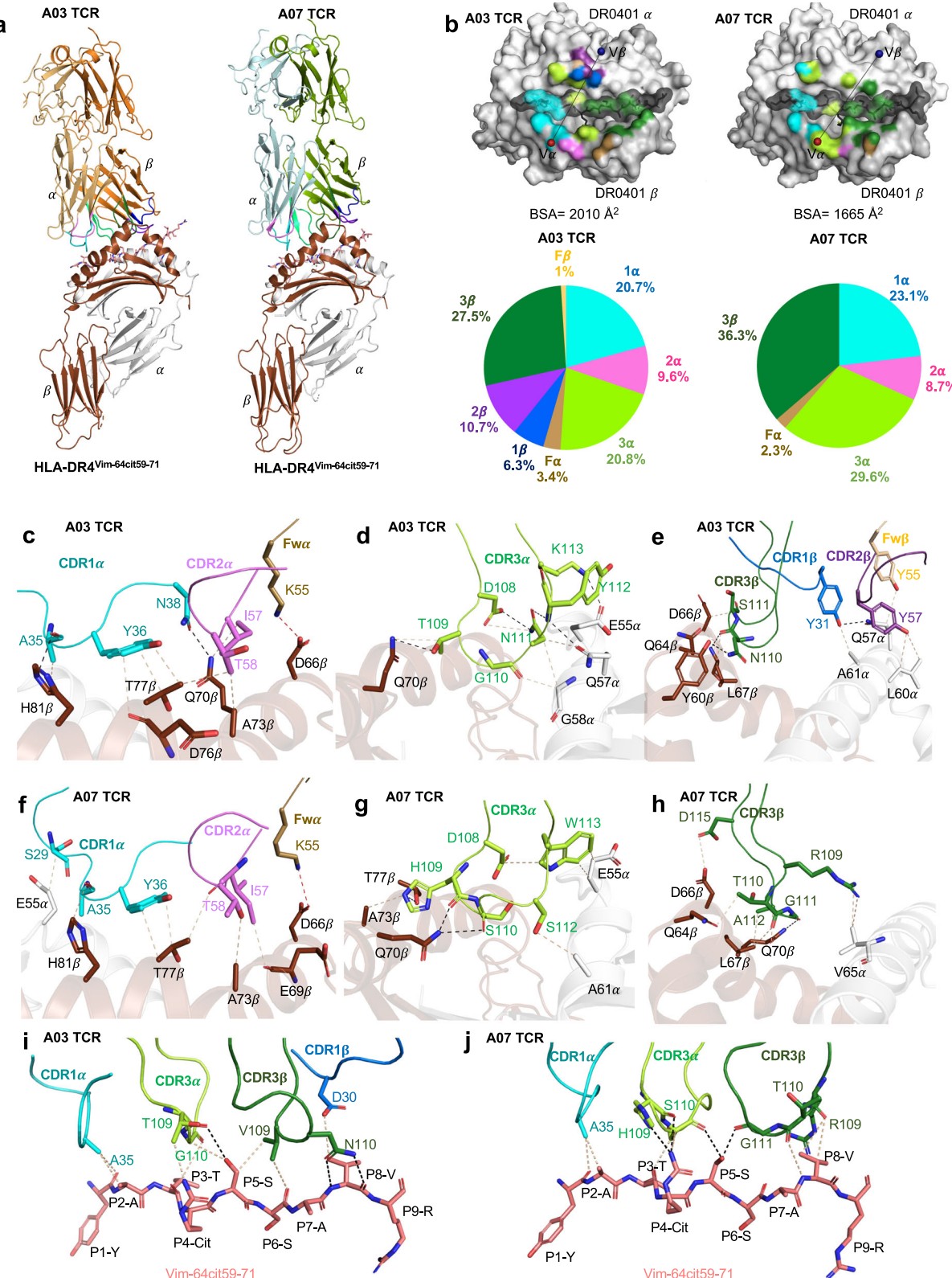

**Fig. 4 | TRAV6⁺ TCR recognition of HLA-DR4^Vim-64cit59-71.** **a** Overall cartoon representation of immune TCR A03 and A07 complexed with HLA-DR4^Vim-64cit59-71. The HLA-DR4 α and β chains are coloured in white and brown, respectively. The peptide is coloured in pink sticks. The CDR loops 1α, 2α, and 3α are highlighted in cyan, violet, and light green colour, whereas 1β, 2β, 3β are coloured in blue, purple, and dark green, respectively. The FW α residues are coloured in sand and β residues are coloured in beige. **b** Top: Surface representation of TCR footprint on pHLA with A03 TCR (left panel) and A07 TCR (right panel). TCR footprint colours are in accordance with the nearest TCR contact residue. The Vα and Vβ centre of mass positions are shown in red and blue spheres, respectively, and connected via a black line. Bottom: Pie charts represent the relative contribution of each CDR loop and FW residues of TCR to the interface with HLA-DR4^Vim-64cit59-71. Detailed interactions of A03 TCR between (**c**) CDR1α, 2α and FW α, (**d**) CDR3α, (**e**) CDR1β, 2β, 3β and FW β with HLA-DR4, and (**i**) peptide interactions are shown. Detailed interactions of A07 TCR between (**f**) CDR1α, 2α and FW α, (**g**) CDR3α, (**h**) CDR3β with HLA-DR4, and (**j**) peptide are shown.

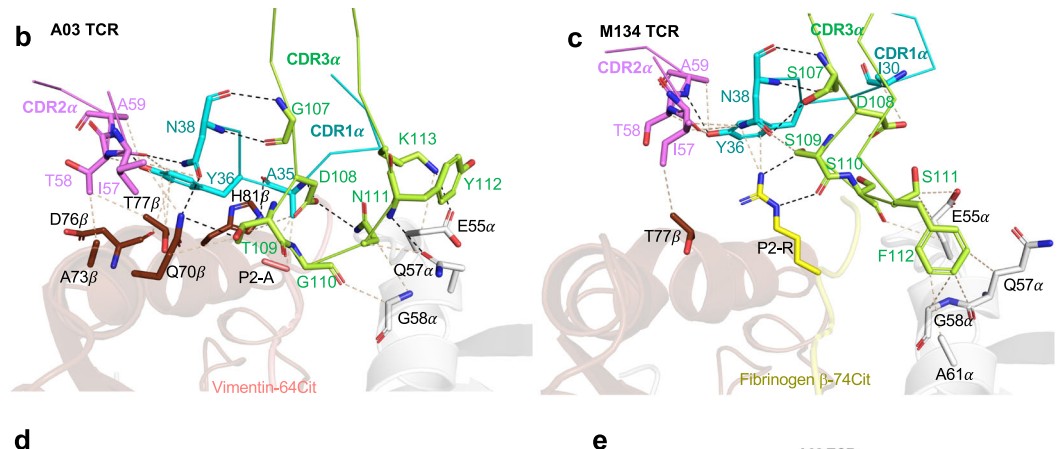

| Clone | TRAV | TRAJ | TRBV | TRBD | TRBJ | CDR1a | CDR2a | CDR3a |
|-------|------|------|------|------|------|-------|-------|-------|
| A03 | 6-6 | 40 | 13-3 | 1 | 1-3 | TTSI....AYPN | VITA...GQK | CALGDTGNYKY |
| M134 | 6D-6 | 50 | 4 | 2 | 2-3 | ATSI....AYPN | VITA...GQK | CALSDSSSFSKL |

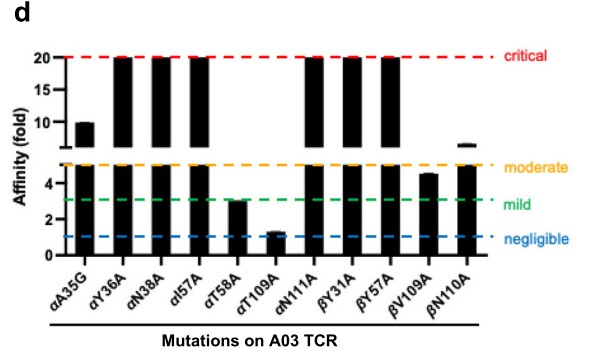

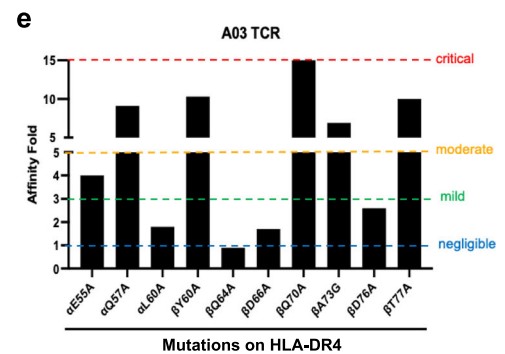

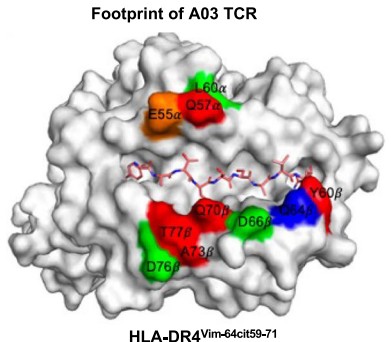

**Fig. 5 | Bias of TRAV6+ TCR recognition toward HLA-DR4^Vim-64cit59-71 and HLA-DR4^Fibβ-74cit69-81. a** TRAV CDR loop sequences of M134 TCR and A03 TCR. Detailed interactions between CDR α loops of TRAV6+ (**b**) A03 TCR and (**c**) M134 TCR docking on top of the antigen binding cleft at P2 of the epitope. The HLA-DR4 α- and β-chains are coloured in white and brown, respectively, Vim-64cit59-71 peptide coloured is coloured in pink and Fibrinogen β-74cit69-81 peptide is coloured in yellow. **d** Effect of A03 TCR point mutations at the pHLA interface. The Y-axis represents the fold of affinity of mutant TCRs as compared with wild-type TCR. The X-axis represents the position of A03 TCR mutants. **e** Effect of HLA-DR4 mutations

on A03 TCR binding affinity. The Y-axis represents the fold change in affinity of mutant TCRs with respect to the wild type A03 TCR and the X-axis represents the position of HLA-DR4 residues mutation. The SPR experiments were performed in duplicate (n = 2). The impact of each mutation was classified as negligible (≤ 1-fold affinity decrease, blue), mild (1.5-fold to 3-fold affinity decrease, green), moderate (3-fold to 5-fold affinity reduction, orange), or critical (>5-fold affinity decrease or no binding, red) shown on graph and on surface of pHLA. Source data are provided as a Source Data file.

shown in Fig. 5b, c. Alanine substitution of CDR2 Thr58α had a moderate effect on HLA-DR4 recognition as this residue only involved HLA-DR4 interaction. As such, the point mutagenesis study of germline encoded residues have corresponded with our current and previously defined structural data, as shown in Vim-64cit59-71 restricted A03 TCR

and Fibβ-74cit69-81 restricted M134 TCR (Fig. 5b, c), respectively, highlighting the role of these conserved TRAV6+ residues in maintaining the TCR inter-CDRα loop contacts and peptide-HLA-DR4 recognition. Alanine substitution of CDR3 Thr109α had no effect on the affinity of the interaction, consistent with the prediction from the

structural data that an alanine mutation could maintain the interaction with Gln[70]β in the SE (Fig. 4d). On the other hand, CDR3α Asn[111] to alanine mutation had a critical impact on pHLA recognition, which is indicative of the role of Asn[111] residue in maintaining the brace-like docking of the adjacent CDR3α residues in the antigen binding cleft, thereby allowing extensive interactions with HLA-DR4 (Figs. 4d, 5b). Moreover, mutation in the A03 TCR β-chain (CDR1β Tyr[31], CDR2β Tyr[57], and CDR3β Asn[110]) critically impacted the interactions to HLA-DR4[Vim-64cit59-71] (Fig. 5d, Supplementary Fig. 3). Together, the structural details and determination of energetically critical residues of the conserved *TRAV6*[+] encoded TCRs, highlighted the importance of germline encoded CDRα residues, as well as non-germline encoded CDR3α and 3β residues for both HLA-DR4 and Vim-64cit[59-71] peptide recognition.

To further investigate which residues on HLA-DR4 molecule drive biased selection of *TRAV6*[+] genes in the Vim-64cit[59-71] peptide reactive CD4[+] T cell repertoire, we made point mutations on the HLA-DR4 molecule based on structural contacts observed in A03 TCR-pHLA ternary complex and undertook SPR analysis using the A03 TCR and mapped the critical interactions to the surface of HLA-DR4[Vim-64cit59-71] (Fig. 5e). Consistent with our previous study on the M134 TCR-pHLA complex[30], the SE residues Gln[70]β and Ala[73]β played critical roles in TCR recognition (Fig. 5e, Supplementary Fig. 4). Among the conserved contact residues Glu[55]α, Gln[57]α, Gly[58]α, and Thr[77]β of HLA-DR4 within both the A03 and M134 TCRs in pHLA ternary complexes (Fig. 5b, c), alanine mutation to Gln[57]α and Thr[77]β showed ~10-fold reduced affinity compared to wildtype HLA-DR4 recognition while Glu[55]α showed ~4-fold reduced affinity after mutation (Fig. 5e, Supplementary Fig. 4). Accordingly, point mutation of HLA-DR4 Glu[55]α, Gln[57]α, and Thr[77]β residues further confirmed their role on *TRAV6*[+] gene selection.

## TCR recognition of HLA-DR4[α-eno-15cit10-22V20G]

We next determined the basis of TRAV26-1 TRBV2[+] RA2.7 TCR recognition of HLA-DR4[α-eno-15cit10-22V20G]. We solved the structure of the RA2.7 TCR in complex with HLA-DR4[α-eno-15cit10-22V20G] to a resolution of 2.4 Å (Fig. 6a, Supplementary Table 2). The RA2.7 TCR docked ~ 60° across the central region of the antigen binding cleft, with a total BSA of ~ 1940 Å[2]. The RA2.7 TCR Vβ-chain dominated the pHLA interaction, with 62% of the total BSA, compared to 38% conferred by the Vα-chain (Fig. 6a). In the RA2.7 TCR, the CDR3β loop served as the major contributor to interactions with HLA-DR4[α-eno-15cit10-22V20G], with 33% BSA compared to ~17%, 7%, and 5% for CDR2β, CDR1β loops and FWβ regions, respectively, and the CDR1α and 3α loops contributing ~ 17% BSA each (Fig. 6a, Supplementary Table 3). Unlike the TRAV6[+] CDR1α which located on top of the peptide, CDR1α of TRAV26-1 was positioned on top of HLA-DR4 β-chain and extensively interacted with Gln[70]β, Ala[73]β, Thr[77]β and His[81]β (Fig.6b, Supplementary Table 6). The CDR2α loop, which only contributed 3% BSA, was peripherally located from the peptide-binding groove and contacted Ala[73]β and Thr[77]β of HLA-DR4 with Leu[57]α. The CDR3α loop transversed the peptide-binding cleft and interacted with the HLA-DR4 α-chain (Gly[58]α and Asn[62]α) and β-chain (Gln[70]β) via Asn[110]α and Ala[109]α, respectively (Fig. 6b). The HLA-DR4 α-chain was mainly contacted by the CDR1β, 2β loops and the FWβ region (Fig. 6c). Tyr[57] in CDR2β had multiple contacts with Ala[64], Val[65], and Ala[68] residues in HLA-DR4α. Moreover, Asn[58] and Glu[64] in CDR2β also mediated a vdW and salt bridge interaction with HLA-DRα Ala[68] and Lys[67], respectively (Fig. 6c). The RA2.7 TCR CDR3β sat on top of HLA-DR4 β-chain and was stabilized by Arg[108]β and Tyr[113]β forming hydrogen bonds with Tyr[60]β, Gln[64]β and Asp[66]β from the HLA-DR4 β-chain. The Tyr[110]β of CDR3β pointed toward the peptide binding groove forming multiple hydrogen bonds with SE Gln[70]β, Lys[71]β as well as P4-Cit of the peptide, while CDR3 Phe[111]β further stabilized the CDR3β through extensive interactions with HLA-DR4 Gln[64]β, Leu[67]β, and Gln[70]β (Fig. 6d).

At the RA2.7 TCR-HLA-DR4[α-eno-15cit10-22V20G] complex interface, Asn[36] in CDR1α formed a hydrogen bond and vdW interactions with P2-Asp,

of the peptide, while Pro[108] and Asn[110] in CDR3α interacted with P2-Asp, P3-Ser and P5-Gly (Fig. 6e). [109]DYF[111] residues in the CDR3β loop on the other hand were pointed downward into the antigen binding cleft and contacted the side chain of the P4-Cit, as well as main chain interactions with P6-Asn, P7-Pro and P8-Thr. Furthermore, the Leu[37] and Tyr[57] in CDR1β and 2β contacted P8-Thr via vdW interactions and hydrogen bonds, respectively.

To investigate the determinants governing human TRAV26-1-TRBV2[+] TCR recognition of HLA-DR4[α-eno-15cit10-22V20G], we performed alanine-scanning mutagenesis on interface residues of the RA2.7 TCR and measured the impact of the mutants via SPR (Fig. 6f, Supplementary Fig. 5). Alanine substitution at residues CDR1α (Asn[36], Tyr[38]), 2α (Leu[57]), 2β (Tyr[57]), and 3β (Asp[109], Tyr[110], and Phe[111]) of the RA2.7 TCR, which contacted the SE of HLA-DR4 and/or the α-eno-15cit[10-22V20G] peptide, almost completely abolished pHLA interaction (≥ 10-fold affinity reduced compared to wildtype TCR). In contrast, mutations in CDR3α (Pro[108] and Asn[110]), as well as CDR2β (Asn[58] and Glu[64]) had limited impact on the TCR-pHLA affinity (≤ 3-fold affinity reduced compared to wildtype TCR). Moreover, alanine replacement of Arg[108] in CDR3β revealed moderate impact to TCR-pHLA affinity, while mutation at Tyr[113] in CDR3β had no impact on TCR-pHLA interaction.

Thus, despite the RA2.7 *TRBV2* encoded β-chain dominating interactions at the interface with HLA-DR4[α-eno-15cit10-22V20G], crucial energetic contributions to the pHLA interaction were attributable to germline encoded residues of TRAV26-1, which explained the biased selection of this gene segment in the CD4[+] T cell repertoire responding to the α-eno-15cit[10-22V20G] epitope (Fig. 3g, Supplementary Table 1).

## Role of the HLA-DR4 SE in TCR recognition

To understand the role of HLA-DR4 in TCR recognition, we first investigated the SE ([70]QKRAA[74]) interaction with TCRs in the HLA-DR4 Vim-64cit[59-71], α-eno-15cit[10-22], and Fibβ-74cit[69-81] restricted complexes. The interactions between the SE residue Gln[70]β and both the TCR and P4-Cit and Lys[71]β, hydrogen bonding with P4-Cit, are conserved in all structures, as observed previously[30] (Fig. 7a–d). In the HLA-DR4[Vim-64cit59-71] restricted A03 TCR complex, Gln[70]β formed a stable tripartite interaction with Thr[109] in the CDR3α and P4-Cit in vimentin epitope (Fig. 7a). In addition to Gln[70]β, the complex was further stabilized by germline encoded CDR1α (Asn[38]) and CDR2α (Ile[57]) residues. Moreover, Ala[73]β in the SE also formed vdW contact with Thr[58] in CDR2α loop. (Fig. 7a). Another Vim-64cit[59-71] restricted A07 TCR also made very similar SE contacts, as in A03 TCR complex, albeit with distinct residues in CDR3α and 3β loops (Fig. 7b). Here, Gln[70]β in the SE pointed down toward the peptide and formed extensive contacts with main chain [109]HS[110] in CDR3α, as well as Gly[111] in the CDR3β loop (Fig. 7b). Lys[71] remained in contact with P4-Cit, which in turn formed a hydrogen bond with His[109] in the CDR3α loop. Moreover, Ala[73] in the SE also made vdW interactions with CDR2α Thr[58] and CDR3α His[109] of the A07 TCR (Fig. 7b). Thus, for citrullinated vimentin TCRs, our structural findings confirmed that TRAV6[+] CDR loops were critical in SE recognition, as well as the citrulline itself that interacted with the SE and the TCR.

In the RA2.7 TCR-HLA-DR4[α-eno-15cit10-22] complex similar comprehensive interactions were observed, yet interactions between the SE, P4-Cit and CDR loops predominantly involved the TRBV2[+] Vβ-chain. Such TRBV biased recognition of the SE is consistent with our previous reported TCR-HLA-DR4[Fibβ-74cit69-81] complex structure[30] (Fig. 7c, d). Here, Gln[70]β in the SE made multiple interactions with the P4-Cit of the α-eno-15cit[10-22] epitope, Tyr[38] in the CDR1α, Ala[109] in the CDR3α, as well as Phe[111] in the CDR3β of the RA2.7 TCR that could explain the bias that TRAV26-1 plays in SE recognition even with a diminished contribution in pHLA contact due to different epitopes (Fig. 7c). Despite constant contact between Lys[71] and P4-Cit, Lys[71] also made direct hydrogen bond contacts with the CDR3β Tyr[110] of the RA2.7 TCR which is absent in M134 TCR-HLA-DR4[Fibβ-74cit69-81] complex. Accordingly, the CDRβ

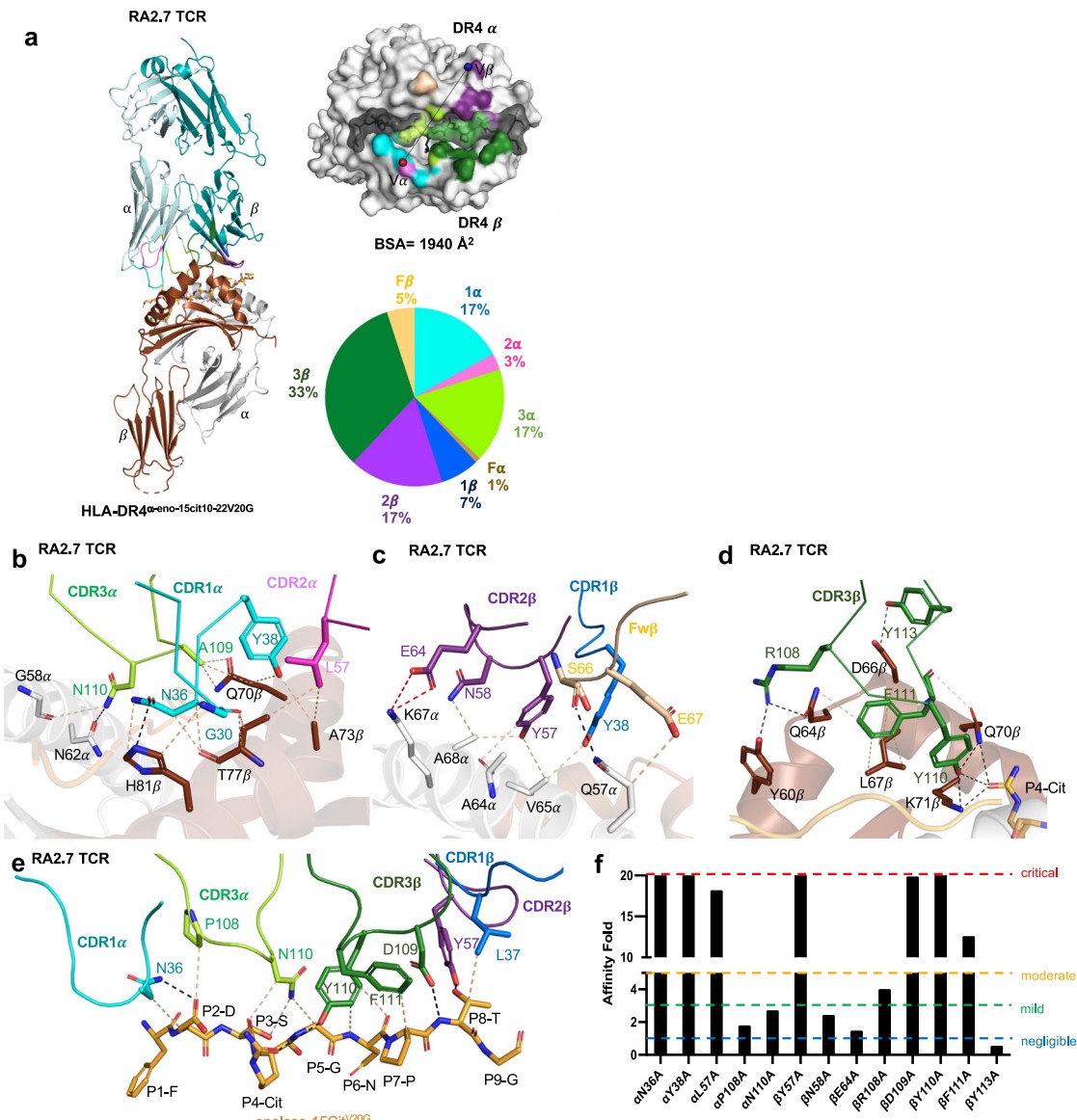

**Fig. 6 | TRAV26-1⁺ TCR recognition of HLA-DR4^α-eno-15cit10-22V20G. a** Left: Overall cartoon representation of human TCR RA2.7 complexed to HLA-DR4^α-eno-15cit10-22V20G. The HLA-DR4 α and β chains are coloured in white and brown, respectively. The peptide is coloured in orange sticks. The CDR loops 1α, 2α, and 3α are highlighted in cyan, violet, and light green colour, whereas 1β, 2β, 3β are coloured in blue, purple, and dark green, respectively. The FW α residues are colour in sand and β residues are colour in beige. Right Top: Surface representation of TCR footprints and TCR docking. TCR footprint colours are in accordance with the nearest TCR contact residue. The Vα and Vβ centre of mass position is shown in red and blue spheres, respectively, and connected via a black line. Right bottom: Pie charts present the relative contribution of each CDR loop and FW of TCR to the surface of

HLA-DR4^α-eno-15cit10-22V20G. Detailed interactions of RA2.7 TCR between (**b**) CDR1α, 2α and 3α, (**c**) CDR1β, 2β, and FW β, **d** CDR3β with HLA-DR4 and (**e**) peptide interactions are shown. **f** Effect of RA2.7 TCR point mutations at the pMHC II interface. The SPR experiments were performed in duplicate (*n* = 2). The Y axis represents the fold of affinity of mutant TCRs as compared with wild-type TCR. The X axis represents the position of RA2.7 TCR mutants. The impact of each mutation was classified as negligible (<1.5-fold affinity decrease, blue), mild (1.5-fold to 3-fold affinity decrease, green), moderate (3-fold to 5-fold affinity reduction, orange), or critical (>5-fold affinity decrease or no binding, red). Source data are provided as a Source Data file.

¹¹⁰YF¹¹¹ encoded TRBV2⁺ RA2.7 TCR is critical in HLA-DR4 shared epitope recognition, mimicking the role of '(L)DW(G)' motif in the Fibβ-74cit₆₉₋₈₁ restricted M134 complex (Fig. 7c, d). Collectively, the *TRAV/TRBV* biased HLA-DR4-citrullinated peptide reactive TCRs is important in SE recognition.

## Specificity determinants of TCR-citrullinated antigen interactions

Next, we compared the structures of HLA-DR4^Vim-64cit59-71, HLA-DR4^α-eno-15cit10-22, and HLA-DR4^Fibβ−74cit69-81 complexed with their specific TCRs. These three complexes were superposed with root-

mean-square deviation (RMSD) to the Cα of HLA-DR4 of ~0.4 Å (Fig. 7e).

Consistent with our previous study, residues at P2, P5, P7 and P8 of the bound peptide were solvent exposed, representing potential TCR contact sites conferring TCR specificity[4]. A distinct feature between the peptide epitopes was observed at P2, where there was a neutral-charged alanine residue in the Vim-64cit₅₉₋₇₁ epitope, a positively charged arginine in Fibβ-74cit₆₉₋₈₁; and a negatively charged aspartate in α-eno-15cit₁₀₋₂₂, which in turn affected the overall surface charge of the pHLA, and thus impacted TCR repertoire selection (Fig. 7e, Supplementary Fig. 6a–h). The overall surface charge of pHLA

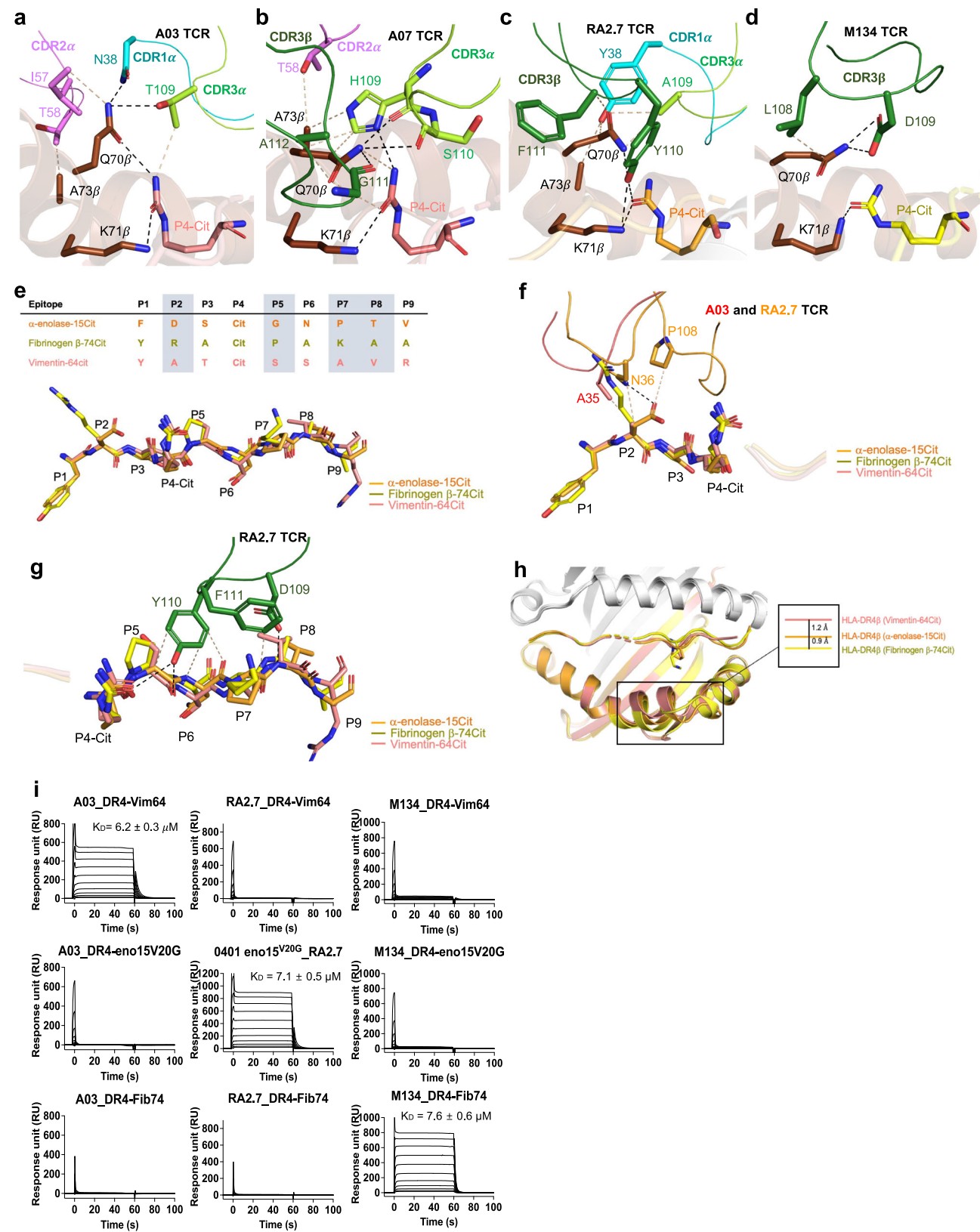

at P2 was a weak negative charged in Vim-64cit$_{59-71}$ epitope, a weak positive charged in arginine in Fibβ−74cit$_{69-81}$; and a strong negative charged in aspartate of α-eno-15cit$_{10-22}$ epitope (Supplementary Fig. 6e−h). The significant difference in the overall surface charge of pHLA determined the specific TCR repertoire for each epitope. The strong negative charged aspartate P2 of α-eno-15cit$_{10-22}$ in the peptide

binding groove disfavours TRAV6⁺ TCRs with Asp108 CDR3 from transverse docking on top of the peptide binding groove due to charged repulsion as observed in the A03, A07 and M134 TCR-pHLA-complexes, which may explain the observation in the α-eno-15cit$_{10-22}$ reactive TCR repertoire of the TRAV6⁺ CDR3 encoded ${}^{108}$GG${}^{109}$ motif (Figs. 1c, 7e, 4d, 4g, Supplementary Fig. 6 g). Besides the residue charge

**Fig. 7 | Antigen specificity on HLA-DR4 towards T cell receptors.** Detailed interactions of SE residues with P4-Cit and (**a**) A03 TCR, (**b**) A07 TCR, (**c**) RA2.7 TCR, and (**d**) M134 TCR. Black dash represents a hydrogen bond and beige dash represent vdW interaction. **e** Top: Peptide sequences of α-eno-15cit$_{10-22}$, Fibβ-74cit$_{69-81}$ and Vim-64cit$_{59-71}$ from pocket 1 to pocket 9. Bottom: The overlay of α-eno-15cit$_{10-22}$ (orange), Fibβ-74cit$_{69-81}$ (yellow), and Vim-64cit$_{59-71}$ (pink) peptides from ternary complex structure. **f** Overlaid N-terminal region (P1-P4 position) of α-eno-15cit$_{10-22}$ (orange), Fibβ-74cit$_{69-81}$ (yellow), and Vim-64cit$_{59-71}$ (pink) peptides and interactions with TCRs (A03 in red, and RA2.7 in orange). **g** Overlaid C-terminal region (P5-P9 position) of α-eno-15cit$_{10-22}$ (orange), Fibβ-74cit$_{69-81}$ (yellow), and Vim-64cit$_{59-71}$

(pink) peptides and interactions with CDR3β of RA2.7 TCR. **h** Overlaid pHLA structures of α-eno-15cit$_{10-22}$ (orange), Fibβ−74cit$_{69-81}$ (yellow), and Vim-64cit$_{59-71}$ (pink) ternary complexes. The deviation in α2 helix of the peptide antigen binding cleft was highlighted and measured (unit in Å). **i** The affinity analysis of A03 TCR, RA2.7 TCR and M134 TCR for HLA-DR4 with Vim-64cit$_{59-71}$, α-eno-15cit$_{10-22V20G}$, and Fibβ−74cit$_{69-81}$ epitopes. All data were derived from three independent measurements with maximal TCR concentrations of 100 $\mu$M ($n=3$). For each concentration the points represent the mean and error bars correspond to SD. RU denotes response units. Source data are provided as a Source Data file.

at P2, the difference in atomic length of the P2 residue also contributes towards TCR selection. The long side chain of the Fibβ-74cit$_{69-81}$ P2-Arg compared to the shorter Vim-64cit$_{59-71}$ P2-Ala and α-eno-15cit$_{10-22}$ P2-Asp, disfavoured the CDR1α of the TCR to come into contact with the pHLA which is required in the Vimentin and α-enolase TCR-pHLA interactions (Fig. 7f).

The combination of the flexible orientation of the SE Gln$^{70}$β with P5, P7 and P8 residues from the antigen further determined TCR specificity where TRBV interactions were dominant (Fig. 7a–e, Supplementary Fig. 6e–h). The P5 of Vim-64cit$_{59-71}$, α-eno-15cit$_{10-22}$, and Fibβ-74cit$_{69-81}$ peptides contained a serine, glycine or proline residue, respectively (Fig. 7e, g). The P5-Ser or P5-Pro in the Vim-64cit$_{59-71}$ and Fibβ-74cit$_{69-81}$ epitope, respectively, prevented the CDR3β $^{109}$DYF$^{111}$ determinant of the RA2.7 TCR from projecting into the antigen binding groove to interact with the SE and epitope (Fig. 7g). Likewise, the Fibβ-74cit$_{69-81}$ epitope P7 Lys prevented the RA2.7 CDR3β $^{109}$DYF$^{111}$ determinant making similar pHLA contacts. Although P7 was not involved in TCR interactions, the side chain of the P7 residue determined the flexibility of the HLA β-chain in the antigen binding cleft spanning residues Gln$^{64}$ to Lys$^{71}$. The P7-Lys in Fibβ-74cit$_{69-81}$ and P7-Pro in α-eno-15cit$_{10-22}$ induces a wider cleft of 2.1 Å and 1.2 Å, respectively, compared to P7-Ala in Vim-64cit$_{59-71}$, thus allowing the CDRβ loops of the Fibβ-74cit$_{69-81}$ and α-eno-15cit$_{10-22}$ reactive TCRs to delve further into the peptide binding cleft making contacts with the pHLA (Fig. 7h). The P7-Lys in Fibβ−74cit$_{69-81}$ also created a positively charged interface that favoured the selection of the DW motif in the Fibβ-74cit$_{69-81}$ reactive T cell repertoire (Supplementary Fig. 6h). In addition, the P8 residue was accommodated by a small residue in all three pHLA complexes, with either a valine, threonine or an alanine in Vim-64cit$_{59-71}$, α-eno-15cit$_{10-22}$, and Fibβ-74cit$_{69-81}$, respectively (Fig. 7e). The P8 residue contributed to TCR selection when the TRBV was dominantly involved in pHLA contacts as observed in the pHLA-TCR complexes for the Fibβ-74cit$_{69-81}$ and α-eno-15cit$_{10-22}$ epitopes. We have further confirmed TCR determinants in citrullinated RA antigen specificity through binding of designated TCRs against citrullinated vimentin, α-enolase, and fibrinogen epitopes using SPR. Notably, corresponding with our structural analysis, no cross-reactivity was observed for these TCRs across all three epitopes (Fig. 7i). Vim-64cit$_{59-71}$-specific A03 TCR, α-eno-15cit$_{10-22}$-specific RA2.7 TCR, and the previously reported Fibβ-74cit$_{69-81}$-specific M134 TCR showed strict specificity toward their respective designated citrullinated epitopes. Overall, the flexibility of the SE Gln$^{70}$β side chain, as well as the epitope features at P2, P5, P7, and P8, were critical for the specific TCR recognition of individual citrullinated epitopes, with no evidence of cross-reactivity.

## Discussion

In this study we examined two different citrullinated epitopes, Vim-64cit$_{59-71}$ and α-eno-15cit$_{10-22}$, where we observed a biased *TRAV6*$^+$ gene usage in T cell repertoires responding to these epitopes. Moreover, this *TRAV6*$^+$ gene usage was also observed in the HLA-DR4$^{Fibβ-74cit69-81}$ restricted CD4$^+$ T cell repertoire previously described[30]. The presence of the same *TRAV6*$^+$ gene bias for three distinct epitopes suggests that this *TRAV6*$^+$ TCR gene usage is preferably binding to HLA-DR4 molecules in the transgenic mouse. Crystal structures revealed

that the TRAV6$^+$ loops CDRα traversed the peptide-binding cleft with the CDR1α loop docking on top of peptides observed in both the Vim-64cit$_{59-71}$ and Fibβ-74cit$_{69-81}$ TCR-pHLA ternary complexes. The distance of TRAV6$^+$ docking on top of pHLA was determined by the P2 residues of citrullinated epitopes which further influenced the dominant contribution of *TRAV* or *TRBV* encoded regions of the TCRs.

In the PBMC of RA donors, we obtained one TCR using the HLA-DR4$^{Vim-64cit59-71}$ tetramer and three TCRs with the HLA-DR4$^{α-eno-15cit10-22}$ tetramer, where all four TCRs showed *TRAV26-1* gene usage across different epitopes. We have previously reported this *TRAV26-1* biased gene selection in Indigenous North Americans with HLA-DRB1*14:02 presenting a citrullinated vimentin epitope[41]. The HLA-DR4 residues that make contact between the RA2.7 TCR and HLA-DR4 are conserved in HLA-DRB1*14:02. Moreover, this *TRAV26-1* gene usage was also observed in HLA-DR4 with citrullinated Tenascin C epitope further suggests that this *TRAV* gene usage selection is driven by HLA-DR4[42]. From the structural studies presented here, CDR1α docked on top of the HLA-DR4 β-chain. Intriguingly, the CDR1α residues (asparagine and tyrosine) either at positions 36 and 38 of TRAV6$^+$ TCRs, or vice versa in TRAV26-1$^+$ TCR, made similar contacts with P2 of peptide and HLA-DR4 β-chain, thus underscoring the relevance of the mouse transgenic model to that of human RA. Similar observations have been found in TCRs utilising the *TRAV26-1* gene in coeliac disease (TCRs S16, D2, JR5.1[43]) and in type I diabetes (TCRs A2.13, A5.5, A5.8[44] and TCRs ET600-1, ET650-1, -2, -3, -4, and -5[45]). Although the HLA-DR4 and HLA-DQ molecules have only ~60% homology, the HLA residues interacting with germline encoded *TRAV26-1* were well conserved (6 out of 7 residues) further supporting that biased *TRAV* gene usage is mainly driven in large part by HLA molecules.

It is difficult to identify the repertoire of citrullinated epitopes recognized by autoreactive T cells in donors with RA because ACPA recognize multiple citrullinated epitopes[46]. There are several well-known candidate citrullinated antigens including citrullinated vimentin (Vim-64cit$_{59-71}$) and α-enolase (α-eno-15cit$_{10-22}$) epitopes that are detected even before disease onset and are associated with RA disease[47]. The pathogenetic mechanisms underlying HLA associations in RA are complex, and the molecular identity of the T cell epitopes and TCRs that contribute to RA are not well-defined, hence studying CD4$^+$ T cells restricted to different antigens presented by HLA-DR4 SE allomorphs in RA provides important information regarding the broader autoimmune response to citrullinated proteins. This would allow us to determine, using biophysical and structural methods, whether antigen specificity is generated to this PTM in the CD4$^+$ T cell response as it has been demonstrated in the ACPA$^+$ humoral response[46]. Indeed, the data we presented demonstrated T cell specificity, with no cross-reactivity, to each citrullinated epitope tested.

Citrullination is responsible for the generation of neo-antigens in RA, whereby not only does it enable antigens to bind to HLA-DR allomorphs having the SE[4,14] but is also involved in TCR recognition[30]. Our results here are consistent with our previous study that the SE has a duality of function in the context of T cell-mediated autoimmunity[30]. However, the interactions between TCRs and the SE will vary based on the features of the citrullinated self-epitopes in question. Positions P2, P5, P7 and P8 of the citrullinated epitopes are solvent exposed in their

respective pHLA complexes and thus represent potential TCR contact points. However, it was the charge characteristic and side chain features of the P2 residues of the peptide that appeared to play an important role in determining the T cell repertoire selected for a given citrullinated neo-antigen, thereby providing a structural basis for TCR specificity to these epitopes as well as the lack of cross-reactivity observed. Overall, our study has provided a deeper understanding of how citrullinated antigens shape CD4$^+$ T cell repertoire specificity.

## Methods

### Ethics
Animal experiments were approved and conducted under guidelines set by the Monash University Animal Ethics Committee. Human experimental work was conducted according to the Australian National Health and Medical Research Council (MHMRC) Code of Practice. Use of PBMC samples for analysis of epitope-specific cells from humans was approved by the Monash University Human Research Ethics Committee (HREC 23019).

### Animals
Transgenic mice expressing HLA-DR4 (*HLA-DRA1*01:01/DRB1*04:01*) on a mouse MHC class II knockout background were purchased from Taconic Biosciences (model 4149)[48]. The HLA-DR4 molecules in these mice contain mouse I-E$^d$ α2 and β2 domains to maintain interaction with the CD4 co-receptor. HLA-DR4 mice were housed in the animal facility at Monash University under the following conditions: ambient temperature 20–22 °C, humidity control 40–70%, 12 h light/12 h dark cycle. Both mouse sexes were used in this study and no differences were observed between them. Mice were euthanized by carbon dioxide asphyxiation.

### Peptides
Peptides used were: 13-mer Vimentin-64cit$_{59-71}$ ($^{59}$GV**YAT**cit**SSAVR**LR$^{71}$; denoted as Vim-64cit$_{59-71}$ where cit represented native arginine residue modified to citrulline), native α-enolase-15cit$_{10-22}$ ($^{10}$EI**FDS**cit**GNP**TVEV$^{22}$; denoted as α-eno-15cit$_{10-22}$), V20G mutant α-enolase-15cit$_{10-22}$ (denoted as α-eno-15cit$_{10-22V20G}$), fibrinogen β-74cit$_{69-81}$ ($^{69}$GG**YRA**cit**PAKA**AAT$^{81}$; denoted as Fibβ-74cit$_{69-81}$) were synthesized by GL Biochem (China). The integrity of the peptides was verified by reverse-phase HPLC and mass spectrometry.

### Peptide immunization
Naïve 13–20 week-old HLA-DR4 mice were immunized subcutaneously in each hind-foot hock with either phosphate-buffered saline (PBS) or 100 μg of a 50:50 mix of Vim-64cit$_{59-71}$ and α-eno15cit$_{10-22}$ peptides emulsified 1:1 in Freund's complete adjuvant (CFA) supplemented with *Mycobacterium tuberculosis* (1 mg/ml) in a total volume of 100 μl (50 μl per hock). Draining LNs (inguinal and popliteal) were harvested for analysis 8 days later. The experiment was performed twice with one PBS-immunized and two peptide-immunized mice per experiment.

### Tetramer-based magnetic enrichment of epitope-specific CD4$^+$ T cells from mice
Tetramer-based magnetic enrichment was used for identification of epitope-specific CD4$^+$ T cells[38,39] in HLA-DR4 mice. Single-cell suspensions of draining LNs (inguinal and popliteal) from d8-immunized HLA-DR4 mice were resuspended in Fc block (2.4G2 supernatant/1% normal mouse serum/1% normal rat serum) with 10 μg/ml phycoerythrin (PE)-labelled HLA-DR4$^{a-eno-15cit10-22}$ and allophycocyanin (APC)-labelled HLA-DR4$^{Vim-64cit59-71}$ tetramers for 1 h at room temperature. Cells were washed once with cold sorter buffer (PBS/0.5% BSA/2 mM EDTA), then resuspended in 400 μl buffer plus 50 μl of each of anti-PE- and anti-APC- conjugated magnetic microbeads (Miltenyi Biotec, #130-048-801 and 130-090-855) and incubated for 30 min at 4 °C. Cells were washed

twice in sorter buffer, resuspended in 3 ml buffer, and passed over a magnetic LS column (Miltenyi Biotec, #130-042-401) according to manufacturer's instructions. The initial flow-through was passed over the column twice, followed by 3 × 3 ml washes with buffer. The column was then removed from the magnet and bound cells eluted by pushing 5 ml of sorter buffer through the column. The eluted cells were then incubated for 30 min at 4 °C with a cocktail of conjugated antibodies to identify epitope-specific cells from immune mouse CD4$^+$ T cell populations (B220, F4/80, NK1.1, TCRβ, CD3, CD62L, CD8, CD4, and viability stain Live/Dead Fixable Aqua; listed in Supplementary Table 7). Cells were finally washed and entire samples (including two rinses of sample tubes) were acquired on a FACSAria III cell sorter with FACSDiva 8.0.1 software (BD Immunocytometry Systems) following the gating strategy in Supplementary Fig. 1a. Collected data were analysed using Flowjo v10.9.0 (FlowJo). Live B220$^-$F4/80$^-$NK1.1$^-$TCRβ$^+$CD62$^{lo}$CD8$^-$CD4$^+$ HLA-DR4$^{a-eno-15cit10-22}$ and HLA-DR4$^{Vim-64cit59-71}$ tetramer binding cells were single-cell index sorted into 96-well polymerase chain reaction (PCR) plates (Eppendorf #00301286) and stored at -80 °C until use. Frequencies of epitope specific to total CD4$^+$ T cells were determined by ratio of total epitope-specific CD4$^+$ T cell numbers to total CD4$^+$ T cells from unenriched samples.

### RA donors and healthy control donors
Cryopreserved human PBMC samples were obtained from companies that provide high-quality biological specimens (BioIVT, USA and All-Cells, USA) and were provided based on the consent of individual donors for their donation to be utilised in biomedical research. Donors included RA donor 2, a 66 year old male (*HLA-DRB1*04:01, DRB1*11:01*), and RA donor 3, a 93 years old female (ACPA$^+$; *HLA-DRB1*04:01, DRB1*13:03*). Healthy donors 10195 and 4737 were 30- and 27 years old males (*HLA-DRB1*04:01, DRB1*04:01*), respectively. Sex and gender based analyses were not considered in the study design.

### Tetramer-based magnetic enrichment of epitope specific CD4$^+$ T cells in humans
Tetramer-based magnetic enrichment was used for identification of epitope-specific CD4$^+$ T cells in PBMC isolated from two donors with *HLA-DRA1*01:01/DRB1*04:01*$^+$ RA or from two HLA-DRB4$^+$ healthy donors. Briefly, cryopreserved PBMC were thawed and rested overnight in complete medium [RPMI1640 (Invitrogen, #21870), 10% fetal bovine serum (FBS), Hepes, L-glutamine, and penicillin-streptomycin] at 37 °C, 5% CO$_2$. Cells were counted and 22–27 million (RA donor 2), 12 million (RA donor 3) or 50 million (healthy donor) PBMC were resuspended in 60 μl/1 ×10$^7$ cells of sorter buffer (PBS/0.5% BSA/2 mM EDTA) and 20 μl/1 × 10$^7$ cells of anti-human FcR blocking reagent (Miltenyi Biotec, #130-059-901) plus 8 μl/1 × 10$^7$ cells of 500 nM dasatinib (Cell Signaling Technology, #9052 S; final concentration 50 nM) and incubated for 30 min at 37 °C, before the addition of PE- or APC-labelled tetramers (at 10 μg/ml final concentration) for 1 h at room temperature. Cells were then washed and labelled with anti-PE or anti-APC conjugated magnetic microbeads, and tetramer-bound cells were enriched over a magnetic LS column as described above for 'Tetramer-based magnetic enrichment of epitope-specific CD4$^+$ T cells from mice'. Enriched cells were then stained with a cocktail of conjugated antibodies to identify epitope-specific cells from naïve CD4$^+$ T cell populations (CD14, CD19, CD3, CD4 and viability stains Live/Dead Fixable Near-IR or FVS700; listed in Supplementary Table 7). Entire samples (including two rinses of sample tubes) were acquired on a FACSAria III cell sorter with FACSDiva 8.0.1 software (BD Immunocytometry Systems) following the gating strategy in Fig. S1B. Live CD19$^-$CD14$^-$ CD3$^+$ CD4$^+$ HLA-DR4$^{a-eno-15cit10-22}$ and/or HLA-DR4$^{Vim-64cit59-71}$ tetramer binding cells were single-cell index sorted into 96-well polymerase chain reaction (PCR) plates (Eppendorf #00301286) and stored at -80 °C until use.

## Analysis of epitope-specific T cell repertoires

Individual HLA-DR4[Vim-64cit59-71] and HLA-DR4[α-eno-15cit10-22] specific CD4[+] T cells were sorted from immunized HLA-DR4 mice. For human analyses, individual HLA-DR4[Vim-64cit59-71], HLA-DR4[α-eno-15cit10-22] and HLA-DR4[α-eno-15cit10-22V20G] specific CD4[+] T cells were sorted from donors with HLA-DR4[+] RA or healthy controls. For both mouse and human samples, mRNA was reverse-transcribed in $2.5\,\mu l$ using the SuperScript™ VILO™ cDNA Synthesis Kit (ThermoFisher Scientific, # 11754) (containing 1x VILO™ reaction mix, 1 x Super-Script™ enzyme mix, 0.1% Triton X-100), and incubated at $25\,°C$ for 10 min, $42\,°C$ for 120 min and $85\,°C$ for 5 min. The entire volume was then used in a $25\,\mu l$ first-round PCR reaction with 1.5 U Taq DNA polymerase (Qiagen, #20120), 1x PCR buffer, 1.5 mM $MgCl_2$, 0.25 mM dNTPs and a mix of 25 mouse or 40 human TRAV external sense primers and a mouse or human TRAC external antisense primer, along with 19 mouse or 28 human TRBV external sense primers and a mouse or human TRBC external antisense primer[49,50] (Supplementary Tables 8 and 9; each at 5 pmol/µl), using standard PCR conditions. For the second-round nested PCR, a $2.5\,\mu l$ aliquot of the first-round PCR product was used in separate TRAV- and TRBV-specific PCRs, using the same reaction mix described above; however, a set of 25 mouse or 40 human TRAV internal sense primers and a mouse or human TRAC internal antisense primer, or a set of 19 mouse or 28 human TRBV internal sense primers and a mouse or human TRBV internal antisense primer, were used (Supplementary Tables 8 and 9)[49,50]. Second-round PCR products were visualized on a gel and positive reactions were purified with ExoSAP-IT reagent (ThermoFisher Scientific, #78201). Purified products were used as template in sequencing reactions with mouse or human internal TRAC or TRBC antisense primers (Supplementary Tables 8 and 9) and sequenced on an ABI 3730 DNA Sequencer at the Monash Micromon Genomics Facility (Monash University), and *TRBV* and *TRAV* gene usage was determined using IMGT/V-QUEST[51]. Selected P2A-linked TCRαβ gene constructs were custom ordered from Genscript and cloned into pMIGII (RRID: Addgene_52107; a gift from D.A.A. Vignali) vector and sequenced to confirm the correct TCR sequence.

## In vitro TCR expression

Human embryonic kidney (HEK) 293 T cells (ATCC, #CRL-3216) were maintained in a humidified incubator at $37\,°C$ and 10% $CO_2$. HEK293T cells were plated at $3.5 \times 10^5$ cells/ well of a six-well plate in 3.5 ml of complete medium [Dulbecco's modified Eagle's medium (DMEM; Invitrogen, #11960), 10% fetal bovine serum (FBS), Hepes, L-glutamine, and penicillin-streptomycin]. The following day, $4.2\,\mu l$ of FuGene 6 HD (Promega, #E2691) was added to $171\,\mu l$ of OptiMEM (Invitrogen, #31985) in an Eppendorf tube and incubated for 10 min at room temperature (RT). The FuGene: OptiMEM mixture was then added dropwise to 700 ng of pMIGII encoding an αβTCR sequence and 700 ng of pMIGII encoding CD3γδε and ζ subunits[52] and incubated for a further 15 min at RT. The FuGene-OptiMEM-DNA mixture was then added dropwise to a well(s) of 293 T cells in the six-well plate and swirled to mix gently before returning to the incubator. After 48 h, the culture medium was aspirated and cells were detached from the plate by repeated washing with fluorescence-activated cell sorting (FACS) buffer [phosphate-buffered saline (PBS) + 0.1% bovine serum albumin]. Transfected 293 T cells were labelled with indicated HLA-DR4 tetramers for 1 h at RT, followed by APC conjugated anti-mouse TCRβ or anti-human CD3 antibody (Biolegend) and Live/Dead Aqua Blue viability stain (Supplementary Table 7). Cells were analysed on a BD LSRFortessa™ X-20 with FACSDiva 8.0.1 software (BD Immunocytometry Systems). Collected data were analysed using Flowjo v10.9.0 (FlowJo).

## Protein expression and purification

αβTCRs were designed with extracellular portion of human *TRBC* constant regions and the *TRAV/TRBV* variable human (for human TCRs) or mouse (for mouse TCRs) regions with an engineered disulphide linkage in the constant domains essentially as previously described[53]. Briefly, TCR α- and β-chains were expressed separately in *Escherichia coli* BL21 (DE3). inclusion bodies purified and refolded in buffer containing 100 mM Tris pH8.0, 5 M Urea, 0.4 M L-Arginine, 2 mM EDTA, 0.2 mM PMSF, 0.5 mM oxidised glutathione and 5 mM reduced glutathione for 48 -72 h, at $4\,°C$ with rapid stirring. The samples were dialyzed extensively in 10 mM Tris pH8.0 and purified on a DEAE (Cytiva) anion exchange column, followed by size exclusion (Superdex 200, 16/600; Cytiva), hydrophobic interaction (Hi Trap SP HP; Cytiva) and anion exchange (HiTrap Q HP column; Cytiva) chromatography.

The extracellular domains of the α- and β-chains of HLA-DR4 (*HLA-DRA*01:01* and *HLA-DRB1*04:01*) were cloned in the lentiviral vectors pLV-EF1α-MCS-IRES-GFP and pLV-EF1α-MCS-IRES-RFP (Biosettia), respectively and HLA-DR4 lentivirus produced through co-transfection of these vectors along with viral packaging plasmids (pMD2.G, pMDLg/pRRE, pRSV-REV; Addgene) in HEK293T cells following the manufacturer's (Biosettia) protocol. Virus transduction was performed in HEK293S (GnTI) (CRL-3022, ATCC) cells stably expressing HLA-DR4 were sorted by single cell FACS (Becton Dickinson). Stably expressed clones scaled up for expression in Expi293 Expression Medium (serum free media; Gibco, Thermo Fisher Scientific) in shaking flasks (120 rpm) at $37\,°C$ in 5% $CO_2$. HLA-DR4 protein was purified from cell culture supernatant as previous described[4]. Briefly, the soluble HLA-DR4 protein was purified from the cell culture supernatant via concentration and buffer exchange (10 mM Tris pH8.0, 500 mM NaCl) using tangential flow filtration (TFF) on a Cogent M1 TFF system (Merck Millipore), followed by immobilized metal ion affinity (Nickel-Sepharose 6 Fast Flow; Cytiva), and size exclusion (Superdex 200, 16/600; Cytiva) chromatography.

## T cell stimulation assay

The RA2.7, A03 or A07 TCR was transduced into the SKW3 T-cell line (ACC 53, sourced from the German Collection of Microorganisms and Cell Cultures (DSMZ)) for stable expression using the lentiviral transduction system as above mentioned. SKW3-transduced RA2.7 T cells or A03 T cells or A07 T cells and BLCL 9031 (*HLA-DRA*0101, HLA-DRB1*0401*; sourced from The International Histocompatibility Working Group (IHWG) Cell and DNA Bank) cells were cultured in 96-well cell culture plates at $37\,°C$ in 5% $CO_2$ in RPMI 1640 media (Gibco, Thermo Fisher Scientific) with 10% Foetal Calf Serum (FCS; Merck). Serial dilution of synthetic peptides of native α-eno-15cit[10-22] peptide or mutant α-eno15cit[10-22V20G] peptide or vim64cit[59-71] peptide, or with anti-DR4 antibody (clone LB3.1, blocking antibody; negative control) was added to $0.1 \times 10^6$ BLCL 9031 cells and incubated for 4 h at $37\,°C$ in 5% $CO_2$. Next, $0.1 \times 10^6$ SKW-3 RA2.7 or A03 or A07 TCR cells or untransduced SKW-3 parental cells (TCR deficient; German Collection of Microorganisms and Cell Cultures; negative control) were added to each well accordingly and incubate overnight at $37\,°C$ in 5% $CO_2$. The cells were then washed twice by centrifugation (350 g for 5 min) and resuspension in FACS buffer (Phosphate buffered saline, containing 10% FCS) then stained a mixture of 1:100 diluted with BUV395 mouse anti-human CD3 (clone UCHT1, BD Biosciences) and APC Mouse Anti-Human CD69 (clone FN50, BD Biosciences) for 1 h on ice in the dark. The cells were washed twice by centrifugation (350 g for 5 min) and resuspension with PBS buffer to remove excess antibodies followed by live/dead cells staining with Zombie NIR (Biolegend) for 30 min at $20\,°C$ in the dark. After live/dead cells staining, the samples were washed 5 times by centrifugation (350 g for 5 min) and resuspension with FACS buffer and subsequently analysed via flow cytometry

(BD LSRFortessa™ X-20). Collected data were analysed using Flowjo v10.9.0, and plotted with GraphPad Prism-9. Three or five independent experiments were conducted. Statistical significance was determined using one way ANOVA multiple comparison of the MFI of unstimulated SKW-3 TCR versus the peptide-stimulated SKW-3 TCR cells.

## Peptide loading of HLA-DR4

HLA-DR4-CLIP was digested with Factor Xa (New England BioLabs, MA, USA) to cleave the covalently linked Strep-TactinII-CLIP peptide. A 20-molar excess of peptide was loaded into the StrepClip-Factor Xa cleaved HLA-DR4 in buffer comprising 50 mM trisodium citrate pH5.4 and 5 mM EDTA at 37 °C for 24 h with HLA-DM as catalysed. The peptide loaded HLA-DR4 was separated from unloaded HLA-DR4-StrepClip using Strep-Tactin Sepharose (IBA, Gottingen, Germany). The monomeric HLA-DR4 complexes were subjected to HRV 3 C protease cleavage at 4 °C for 24 h to remove the fos/jun leucine zippers and further purified using anion exchange (HiTrap Q HP column: Cytiva) chromatography. Target fractions were pooled, buffer exchanged into 10 mM Tris pH8.0, 150 mM NaCl, and concentrated.

## Crystallization, data collection and processing

TCR-HLA-DR4$^{Vim-64cit59-71/\alpha-eno-15cit10-22V20G}$ ternary complexes were concentrated to 10 mg/ml and broad screen crystallisation trials were undertaken at the Monash Molecular Crystallisation Platform using an automated robotic NT8 system. Conditions yielding crystals were further upscaled and optimised via hanging-drop vapor diffusion method in 24 well plates. The A03 ternary complex crystals were obtained in conditions containing 0.1 M Tris pH7.5 – pH8.5, 0.2 M ammonium sulfate, and 20% w/v PEG8K; the A07 ternary complex was crystallized in solutions containing 0.2 – 0.3 M Disodium Malonate, 20%w/v PEG3350, 0.1 M BTP pH6.5 to pH6.9, and 0.3 M triglycine as additive; the RA2.7 ternary complex was crystallized in 4–10% Tacsi-mate and 18-23% w/v PEG3350. Protein was mixed at 1:1 ratio with each crystal condition ("mother liquor") and equilibrated against 500 μl of mother liquor. Crystals were cryoprotected in the mother liquor well solution supplemented with 20% glycerol prior to flash freezing in liquid N$_2$. Diffraction data was collected on the MX2 Beamline of the Australian Synchrotron, using an Eiger x16M detector and processed and scaled with XDS and CCP4 Software Suite, version 8.0.

## Structure determination, refinement, and validation

Complex structures were solved by molecular replacement in PHASER (CCP4 Software Suite, version 8.0) using a separate search model for the TCR (PDB ID: 6V1A) and HLA-DR4 (PDB ID: 4MCZ). Repeated rounds of model building in Coot[54], manual and automated refinement using REFMAC (CCP4 Software Suite, version 8.0) and PhenixRefine (PHENIX[55]) were carried out. The quality of the ternary structures was validated at the Protein Data Bank (PDB) Data validation and deposition services website. The IMGT unique numbering system was used to number TCR variable domains[56]. Data processing and refinement statistics were summarized in Supplementary Table 2. Ramachandran statistic of final models revealed ~96% of residues were in favoured regions, with 0.06-0.19% of residues as outliers. Buried surface area (BSA) calculations were performed using program Areaimol while contact analysis was performed using Contacts (CCP4 Software Suite, version 8.0). All structural figures were generated by PyMOL version 2.5.5.

## Surface plasmon resonance

Affinity measurements were performed using surface plasmon resonance on a Biacore T200 instrument (Cytiva). HLA-DR4 molecules were biotinylated and ~2000 response units was immobilized on a streptavidin (SA) sensor chip (Cytiva). Biotinylated HLA-DR4$^{CLIP}$ was used as a reference cell. Serial dilutions of TCRs were passed over the flow cell surface in 20 mM HEPES pH7.5, 150 mM NaCl, 1 mM EDTA, and

0.005% TWEEN 20 at a flow rate of 10 μl/min. GraphPad Prism v.9.0 (GraphPad Software) was used for data analysis of sensorgrams from which curves were plotted. At least two independent experiments ($n \geq 2$) were performed for each TCR sample. Equilibrium response curves were normalised against the calculated maximum response and the measurements then combined. Data are mean ± standard error of the mean, s.e.m.

## Reporting summary

Further information on research design is available in the Nature Portfolio Reporting Summary linked to this article.

## Data availability

The X-ray crystal structures were deposited in the Protein Data Bank (https://www.rcsb.org/) with the following accession codes: A03 TCR-HLA-DR4$^{Vim-64cit59-71}$, 8TRR; A07 TCR-HLA-DR4$^{Vim-64cit59-71}$, 8TRQ; RA2.7 TCR-HLA-DR4$^{\alpha-eno-15cit10-22V20G}$, 8TRL. All data generated in this study are provided in the Supplementary information/Source Data file or from the corresponding author upon request. Source data are provided as a Source Data file. Source data are provided with this paper.

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

## Acknowledgements

We thank the staff at the Australian Synchrotron for assistance with data collection, the staff at the Monash Macromolecular crystallization facility and the staff at Monash FlowCore. We thank Kylie Loh and Yi Tian Ting for technical assistance. This work was supported by Janssen Pty Ltd. N.L.G. (NHMRC, Project Grant No. 2017335) and J.R. (NHMRC, Project Grant No. 2008981) are supported by National Health and Medical Research Council Investigator awards.

## Author contributions

T.J.L., J.J.L., and C.M.J. performed the research. H.T.D and M.T.T. generated data. D.G.B. provided advice and reagents; T.J.L., J.J.L., C.M.J., N.L.G., H.H.R., and J.R. co-wrote the paper, with editorial input from all co-authors. N.L.G., H.H.R., and J.R. supervised the work.

## Competing interests

This work has been supported by Janssen Research & Development, D.G.B. was an employee of Janssen Research & Development. The other authors declare no competing interests.
