## [Peer Review File · Nature Communications]

The molecular basis underlying T cell specificity towards
citrullinated epitopes presented by HLA-DR4REVIEWER COMMENTS

Reviewer #1 (Remarks to the Author):

In this manuscript, Loh et al perform a structural analysis to investigate features that underlie T cell recognition of citrullinated peptides presented by DRB1*04:01 - the HLA class II allele most strongly associated with anti-citrullinated protein antibodies (ACPA) in rheumatoid arthritis. The authors isolate and study T cell clones specific for two established citrullinated epitopes from proteins that are known to be citrullinated, expressed in the synovium, and recognized by ACPA. Somewhat surprisingly, TCR that recognize the two distinct epitopes share a V alpha preference. The authors utilize expert techniques to draw conclusions about the structural bases for this shared recognition, along with features that are shaped by the individual peptides. The work is relevant and appears well performed, but the authors should note the following points of potential concern:

- 1) In the introduction, the authors cite a limited number of studies to communicate the motivation for their work and choice of antigens and epitopes. Unfortunately, the citations in this section are not all correct (for example a paper on collagen is cited as a CILP reference). Please revisit and correct the citations.
- 2) As noted by the authors, the sequences of murine and human vimentin are not identical, causing a difference at P2 of the studied epitope. It is not clear whether DR4 transgenic mice were immunized with the human or murine sequence. It could be strongly argued that the murine sequence should have been used.
- 3) The authors show that the TCR transductants stain with tetramer, but do all of the cell lines proliferate in response to peptide?
- 4) The observed tetramer frequencies in human subjects are quite low - especially with the wild type peptide, but also with the point mutated peptide. Was something suboptimal with the staining approach? Fresh murine cells were harvested and stained with tetramers immediately. Previously frozen human cells were thawed and given a long overnight rest before staining. Were the cell counts taken before or after resting? If before, it is possible

that an LS column is too large a format for the enrichment. There is a notable age difference between the DR4+ patients and controls - with one RA patient having a very advanced age. Some have reported that antigen specific T cell frequencies decline with age and biologic treatment so these effects could have undermined the success of these staining and enrichment experiments.

5) The authors elected to place a glycine in pocket 9 to generate an improved T cell ligand for DRB*04:01. A previous publication used Serine. Did the authors consider that option for a substitution? In either case the authors should add at least one more comment about the potential influence of using an altered peptide for their remaining experiments.

6) In hindsight, it is clear that the authors have studied two citrullinated epitopes with shared features of recognition. Was there an a priori reason to combine these two?

7) Is the tetramer staining shown in Figure 2 before or after enrichment? Given the low frequencies it would be helpful to verify that enrichment increased the level of observed staining.

8) The overall findings are of interest, but the degree that they can be generalized to other specificities remains opaque.

Eddie James

Reviewer #2 (Remarks to the Author):

Susceptibility to rheumatoid arthritis is correlated with specific HLA class 2 alleles that share a common susceptibility epitope (SE) (Q/R K/R R AA). Autoimmune responses are generated by a group of proteins that undergo post-translational modification in which citrulline is added to specific arginine sidechains leading to charge reversal. These include vimentin, enolase, fibrinogen and tenascin, all of which are present in the synovial fluid of RA patients. It is therefore of importance to understand how the citrullinated peptides (CP) are presented, what structural features in the HLA alleles favour the binding of the CP, how the

citrulline modification stabilises the CPs in the peptide binding cleft and how T-cell receptors are able to recognise the presented CPs but not the unmodified form, enabling the generation of autoimmunity and the production of anti-citrullinated protein humoral response (ACPA). Although partial answers to these questions are known from previous studies the current study provides for the first time a comprehensive understanding at the structural level.

The work focusses on citrullinated epitopes defined in vimentin and α -enolase. CP specific T-cell receptors were produced by immunising humanised HLA-DR4 mice with the vimentin (Fig 4) and enolase (Fig 6) citrullinated self peptides. TCRs were also isolated from human ACPA PBMCs. Biacore experiments found that these TCRs had affinity for HLA within the expected range and crystal structures of the ternary complexes provide a precise molecular explanation for HLA-DR and TCR specificity. Mutagenesis was then used to identify residues important for binding – the rather counter intuitive presentation of this data in Fig 5d,e could be explained more clearly. The authors describe common features that confer TCR specificity in the CPs from vimentin, enolase and the previously characterised peptide from fibrinogen. There are several interactions with conserved residues in the CPs and divergent residues that confer specificity for the A03, RA2.7 and M134 TCRs respectively. Affinity measurements confirm the structural data.

Overall this is a tour de force – the experimental data presented is of high quality and the conclusions drawn compelling. The work will be of interest to a broad range of Nat Comm readers from structural biologists to molecular immunologists and provides a basis for development of therapeutics that target autoimmunity caused by citrullinated neo antigens in rheumatoid arthritis.

Reviewer #3 (Remarks to the Author):

In the manuscript “The molecular basis underlying T cell specificity towards citrullinated peptides presented by HLA-DR4”, Loh et al identify and molecularly characterize multiple TCRs that recognize a series of citrullinated peptides displayed by HLA-DR4. As citrullinated peptides are a primary driver of disease in rheumatoid arthritis, understanding how these epitopes are recognized is relevant for human health. The work here is well done, with a

few outstanding questions about bigger-picture data interpretation.

- A significant portion of this work studies HLA-DR4-restricted TCRs derived from a transgenic mouse. These TCRs largely behave the same as any other TCR, but it is somewhat unclear how much certain features of these TCRs recognizing antigen can directly translate to human TCRs. For example, the observed dominant use of mouse TRAV6 is interesting, but this does not necessarily correspond to a similar mode of recognition or dominant gene usage in human disease. Some discussion about what can – and cannot – be learned in such a system would be helpful for the reader.

- While there is apparent similarity between the A03 and A07 TCR structures and therefore shared features in the TRAV6 gene usage, I have difficulty seeing how the similarity extends to the M134 TCR recognizing the fibrinogen-derived epitope – the contacts look quite different, and indeed the authors state the alpha chain as a whole contributes less to the M134 interface. If TRAV6 is regularly used in these other epitopes, that is interesting – but the ‘why’ of it could be more clearly articulated.

- Moving to the human TCR data, the amino acid differences at P2 along the peptides is interesting, but it should be noted that each of the other TCR contacts differ – it therefore seems overstated to ascribe the specificity differences so strongly to the P2 differences.

- Minor point – it would be good to standardize the boundaries and format for showing CDR loops in Figures 1, 2, and 5.

Reviewers' comments

Reviewer #1 (Remarks to the Author):

In this manuscript, Loh et al perform a structural analysis to investigate features that underlie T cell recognition of citrullinated peptides presented by DRB1*04:01 - the HLA class II allele most strongly associated with anti-citrullinated protein antibodies (ACPA) in rheumatoid arthritis. The authors isolate and study T cell clones specific for two established citrullinated epitopes from proteins that are known to be citrullinated, expressed in the synovium, and recognized by ACPA. Somewhat surprisingly, TCR that recognize the two distinct epitopes share a V alpha preference. The authors utilize expert techniques to draw conclusions about the structural bases for this shared recognition, along with features that are shaped by the individual peptides. The work is relevant and appears well performed, but the authors should note the following points of potential concern:

- 1) In the introduction, the authors cite a limited number of studies to communicate the motivation for their work and choice of antigens and epitopes. Unfortunately, the citations in this section are not all correct (for example a paper on collagen is cited as a CILP reference). Please revisit and correct the citations.
- 2) As noted by the authors, the sequences of murine and human vimentin are not identical, causing a difference at P2 of the studied epitope. It is not clear whether DR4 transgenic mice were immunized with the human or murine sequence. It could be strongly argued that the murine sequence should have been used.
- 3) The authors show that the TCR transductants stain with tetramer, but do all of the cell lines proliferate in response to peptide?
- 4) The observed tetramer frequencies in human subjects are quite low - especially with the wild type peptide, but also with the point mutated peptide. Was something suboptimal with the staining approach? Fresh murine cells were harvested and stained with tetramers immediately. Previously frozen human cells were thawed and given a long overnight rest before staining. Were the cell counts taken before or after resting? If before, it is possible that an LS column is too large a format for the enrichment. There is a notable age difference between the DR4+ patients and controls - with one RA patient having a very advanced age. Some have reported that antigen specific T cell frequencies decline with age and biologic treatment so these effects could have undermined the success of these staining and enrichment experiments.
- 5) The authors elected to place a glycine in pocket 9 to generate an improved T cell ligand for DRB*04:01. A previous publication used Serine. Did the authors consider that option for a substitution? In either case the authors should add at least one more comment about the potential influence of using an altered peptide for their remaining experiments.
- 6) In hindsight, it is clear that the authors have studied two citrullinated epitopes with shared features of recognition. Was there an a priori reason to combine these two?
- 7) Is the tetramer staining shown in Figure 2 before or after enrichment? Given the low frequencies it would be helpful to verify that enrichment increased the level of observed staining.

8) The overall findings are of interest, but the degree that they can be generalized to other specificities remains opaque.

Eddie James

Reviewer #2 (Remarks to the Author):

Susceptibility to rheumatoid arthritis is correlated with specific HLA class 2 alleles that share a common susceptibility epitope (SE) (Q/R K/R R AA). Autoimmune responses are generated by a group of proteins that undergo post-translational modification in which citrulline is added to specific arginine sidechains leading to charge reversal. These include vimentin, enolase, fibrinogen and tenascin, all of which are present in the synovial fluid of RA patients. It is therefore of importance to understand how the citrullinated peptides (CP) are presented, what structural features in the HLA alleles favour the binding of the CP, how the citrulline modification stabilises the CPs in the peptide binding cleft and how T-cell receptors are able to recognise the presented CPs but not the unmodified form, enabling the generation of autoimmunity and the production of anti-citrullinated protein humoral response (ACPA). Although partial answers to these questions are known from previous studies the current study provides for the first time a comprehensive understanding at the structural level.

The work focusses on citrullinated epitopes defined in vimentin and α -enolase. CP specific T-cell receptors were produced by immunising humanised HLA-DR4 mice with the vimentin (Fig 4) and enolase (Fig 6) citrullinated self peptides. TCRs were also isolated from human ACPA PBMCs. Biacore experiments found that these TCRs had affinity for HLA within the expected range and crystal structures of the ternary complexes provide a precise molecular explanation for HLA-DR and TCR specificity. Mutagenesis was then used to identify residues important for binding – the rather counter intuitive presentation of this data in Fig 5d,e could be explained more clearly. The authors describe common features that confer TCR specificity in the CPs from vimentin, enolase and the previously characterised peptide from fibrinogen. There are several interactions with conserved residues in the CPs and divergent residues that confer specificity for the A03, RA2.7 and M134 TCRs respectively. Affinity measurements confirm the structural data.

Overall this is a tour de force – the experimental data presented is of high quality and the conclusions drawn compelling. The work will be of interest to a broad range of Nat Comm readers from structural biologists to molecular immunologists and provides a basis for development of therapeutics that target autoimmunity caused by citrullinated neo antigens in rheumatoid arthritis.

Reviewer #3 (Remarks to the Author):

In the manuscript “The molecular basis underlying T cell specificity towards citrullinated peptides presented by HLA-DR4”, Loh et al identify and molecularly characterize multiple TCRs that recognize a series of citrullinated peptides displayed by HLA-DR4. As citrullinated peptides are a primary driver of disease in rheumatoid arthritis, understanding how these epitopes are recognized is relevant for human health. The work here is well done, with a few outstanding questions about bigger-picture data interpretation.

- A significant portion of this work studies HLA-DR4-restricted TCRs derived from a

transgenic mouse. These TCRs largely behave the same as any other TCR, but it is somewhat unclear how much certain features of these TCRs recognizing antigen can directly translate to human TCRs. For example, the observed dominant use of mouse TRAV6 is interesting, but this does not necessarily correspond to a similar mode of recognition or dominant gene usage in human disease. Some discuss about what can – and cannot – be learned in such a system would be helpful for the reader.

- While there is apparent similarity between the A03 and A07 TCR structures and therefore shared features in the TRAV6 gene usage, I have difficulty seeing how the similarity extends to the M134 TCR recognizing the fibrinogen-derived epitope – the contacts look quite different, and indeed the authors state the alpha chain as a whole contributes less to the M134 interface. If TRAV6 is regularly used in these other epitopes, that is interesting – but the ‘why’ of it could be more clearly articulated.

- Moving to the human TCR data, the amino acid differences at P2 along the peptides is interesting, but it should be noted that each of the other TCR contacts differ – it therefore seems overstated to ascribe the specificity differences so strongly to the P2 differences.

- Minor point – it would be good to standardize the boundaries and format for showing CDR loops in Figures 1, 2, and 5.

Response to reviewers

Reviewer #1

In this manuscript, Loh et al perform a structural analysis to investigate features that underlie T cell recognition of citrullinated peptides presented by DRB1*04:01 - the HLA class II allele most strongly associated with anti-citrullinated protein antibodies (ACPA) in rheumatoid arthritis. The authors isolate and study T cell clones specific for two established citrullinated epitopes from proteins that are known to be citrullinated, expressed in the synovium, and recognized by ACPA. Somewhat surprisingly, TCR that recognize the two distinct epitopes share a V alpha preference. The authors utilize expert techniques to draw conclusions about the structural bases for this shared recognition, along with features that are shaped by the individual peptides. The work is relevant and appears well performed, but the authors should note the following points of potential concern:

We thank the reviewer for their positive appraisal of our manuscript.

Comments:

1) In the introduction, the authors cite a limited number of studies to communicate the motivation for their work and choice of antigens and epitopes. Unfortunately, the citations in this section are not all correct (for example a paper on collagen is cited as a CILP reference). Please revisit and correct the citations.

We thank the reviewer for this correction. We have now revised the Introduction (page 2) as follows:

- Burkhardt H et al., 2005 is now revised to reference for type II collagen and
- we have added ref. 23 James EA et al., 2014 to CILP.

We now state in the revised manuscript on Page 2:

“In RA patients, several proteins found abundantly in the joint synovium and synovial fluid are citrullinated, including vimentin¹⁹, type II collagen²⁰, fibrinogen²¹, α -enolase²², cartilage intermediate layer protein (CILP)²³, and tenascin C^{24, 25}, some of which have been shown to be targets of the immune response^{26, 27}.”

2) As noted by the authors, the sequences of murine and human vimentin are not identical, causing a difference at P2 of the studied epitope. It is not clear whether DR4 transgenic mice were immunized with the human or murine sequence. It could be strongly argued that the murine sequence should have been used.

To clarify, to understand the immune response of RA susceptibility allele HLA-DR4, we utilised a humanised HLA-DRB1*0401-transgenic mouse model that have a mouse MHC class II knockout background but expressed HLA-DRB1*0401. Thus, a human vimentin peptide was chosen in the immunisation study.

We have clarified this point in the revised manuscript (page 3):

“HLA-DR4 transgenic mice were co-immunised with the human Vim-64cit₅₉₋₇₁ and α -eno-15cit₁₀₋₂₂ peptides...”

3) The authors show that the TCR transductants stain with tetramer, but do all of the cell lines proliferate in response to peptide?

Given that the parental cell line used, SKW3, is derived from a T cell leukemia, proliferation can't be used to assess whether a given TCR is capable of transducing a signal as these cells proliferate in the absence of peptide stimulation. To assess whether these TCRs can transduce an activation signal in response to peptide, we produced SKW3 mouse (v-region)-human (constant region) A03 and A07 TCR chimeric constructs and used CD69 upregulation as the readout for stimulation. We now included this data in Supplementary Figure 2 demonstrating that both A03 and A07 chimeric

TCR SKW3 cell lines are activated in response to low concentrations of peptide as measured by CD69 upregulation.

We state (page 5)

“We also generated SKW-3 T cell line transduced with the A03 or A07 mouse-human hybrid TCRs and showed T cell reactivity toward Vim64cit 59-71 peptide at a low peptide concentration (Supplementary Fig. 2a-b).”

4) The observed tetramer frequencies in human subjects are quite low - especially with the wild type peptide, but also with the point mutated peptide. Was something suboptimal with the staining approach? Fresh murine cells were harvested and stained with tetramers immediately. Previously frozen human cells were thawed and given a long overnight rest before staining. Were the cell counts taken before or after resting? If before, it is possible that an LS column is too large a format for the enrichment. There is a notable age difference between the DR4+ patients and controls - with one RA patient having a very advanced age. Some have reported that antigen specific T cell frequencies decline with age and biologic treatment so these effects could have undermined the success of these staining and enrichment experiments.

The tetramer frequencies in RA donor 2 with wildtype α -enol15cit and Vim64cit tetramers were indeed lower than reported in James et al 2014 *Arthritis & Rheumatology* (0.5 and 0.2 per million CD4⁺ T cells respectively compared to an average of 3-4 per million CD4⁺ T cells in James et al). In contrast, tetramer frequency with the point mutant α -enol15V20Gcit peptide was six-fold higher for RA donor 2 (an average of ~2-3 cells per million CD4⁺ T cells in the two RA patients and the two healthy controls), attesting to the more stable interaction of this mutated peptide to HLA-DR4. Importantly, cell counts were always taken after resting and tetramer frequencies calculated based on those counts. Originally, the cell counts listed in the Methods were those reported on the vials prior to freezing. We have now amended these values to report the number of cells after thawing and resting the samples. Our human PBMC samples were obtained cryopreserved and we routinely rest thawed PBMC overnight as this has been reported to reduce the number of apoptotic cells and positively affect lymphocyte functionality (Wang et al 2016 *Cytometry Part A*, 89A; 246-258). We have previously compared tetramer staining of fresh and thawed human PBMC (both rested overnight) from the same donor and obtained similar tetramer frequencies. Note that the numbers of cells listed in Figure 3g were those for which paired TCR $\alpha\beta$ sequence was obtained rather than the number binding the α -enol15 V20Gcit tetramer.

We consider that the lower frequencies we observed are due, in part, to the clean tetramer staining, which minimizes all background, as evidenced by the complete absence of tetramer staining in the CD4- T cell population (Fig. 2A). In addition, the high MFI of the tetramer staining indicates that our low frequencies were not a consequence of poor tetramer labelling. As suggested, it is possible that the tetramer frequencies in this study may also be affected by the advanced age of RA donor 3 and biologic treatment of RA donor 2, whose medications included Actemra (James et al, 2014). Irrespective of tetramer frequencies, we were able to isolate CD4⁺ T cells specific for α -enolase-15cit₁₀₋₂₂ and vimentin-64cit₅₉₋₇₁ and, with these, revealed the basis for biased *TRAV26-1* gene usage in the response.

We have altered the revised manuscript (page 17) to read: “Briefly, cryopreserved PBMC were thawed and rested overnight at 37°C, 5% CO₂. Cells were counted and 22-27 million (RA donor 2), 12 million (RA donor 3) or 50 million (healthy donor) PBMC were treated....”

5) The authors elected to place a glycine in pocket 9 to generate an improved T cell ligand for DRB*04:01. A previous publication used Serine. Did the authors consider that option for a substitution? In either case the authors should add at least one more comment about the potential influence of using an altered peptide for their remaining experiments.

The substitution of P9-Val to P9-Gly Glycine of the α -enolase peptide is based on previous study by our group (Scally et al. 2013. *JEM*), which showed that the binding preference of HLA-DRB1*0401 at position P9 is greatly favoured by a glycine, followed by an alanine and serine residue. To

minimize the potential effect of introducing a non-polar characteristic of wildtype P9 valine residue, a glycine residue was preferred over serine.

We now state in the manuscript (page 5):

“To improve the pHLA stability,...” and our result from activation assay have shown that the altered peptide used did not influence the activation ability of the peptide and SPR proved the stability of antigen improved.

6) In hindsight, it is clear that the authors have studied two citrullinated epitopes with shared features of recognition. Was there an a priori reason to combine these two?

To clarify, we examined these two epitopes (vimentin and enolase) and a previously studied fibrinogen epitope (Lim et al., Sci Immunol. 2021) as:

- they shared the same citrullinated site at position 4
- they have different chemical characteristic in residues exposed to TCR recognition, in particularly P2 (as shown in Figure 7e)
- Citrullinated vimentin, α -enolase, and fibrinogen antibodies are present in the range of 40% to 80% in ACPA-positive RA patients (stated in Page3, with ref. #34-38 in the manuscript).

7) Is the tetramer staining shown in Figure 2 before or after enrichment? Given the low frequencies it would be helpful to verify that enrichment increased the level of observed staining.

Tetramer staining shown in Figure 2 is after enrichment. We have amended the figure legend for Figure 2 in the revised manuscript (page 21) to read: HLA-DR4^{Vim-64cit59-71} and HLA-DR4 ^{α -eno-15cit10-22} tetramer staining on CD4⁺ T cells post tetramer-based magnetic enrichment of PBMC from a HLA-DR4⁺ ACPA⁺ RA patient. Even at previously published frequencies of 3-4 per million specific CD4⁺ T cells (James et al 2014), or higher, it would be impossible to detect true epitope-specific T cells above background in the absence of enrichment. The fact that we are detecting T cells that are genuine epitope-specific cells (as verified by re-expression and tetramer staining) indicates that the enrichment is working.

8) The overall findings are of interest, but the degree that they can be generalized to other specificities remains opaque.

Our understanding of the nature of the T cell repertoire to citrullinated antigens, and molecular bases underpinning this response thereof is germinal. Basic studies, such as that reported here, are required to evaluate the specificity or generality of the findings towards distinct epitopes, which may also help shape our understanding of autoimmune TCR recognition more broadly.

Based on our current and previous findings (Lim et al., Sci Immunol. 2021), three epitopes (citrullinated fibrinogen, vimentin and alpha-enolase) with distinct chemical features give rise to a specific T cell response, while maintaining shared susceptibility epitope recognition. Our comprehensive analysis of TCR-HLA-DR4-citrullinated epitopes complexes at a molecular level deciphered the specific determinants underlying TCR recognition and the preference of TRAV6+ usage in the immune repertoire.

Reviewer #2:

We thank the reviewer #2 for recognising the manuscript as a ‘**tour de force**’ work, ‘**the experimental data presented is of high quality and the conclusions drawn compelling**’, and this work ‘**will be of interest to a broad range of Nat Comm readers from structural biologists to molecular immunologists and provides a basis for development of therapeutics that target autoimmunity caused by citrullinated neo antigens in rheumatoid arthritis**’.

Comment: Mutagenesis was then used to identify residues important for binding – the rather counter intuitive presentation of this data in Fig 5d,e could be explained more clearly. The authors describe common features that confer TCR specificity in the CPs from vimentin, enolase and the previously characterised peptide from fibrinogen. There are several interactions with conserved residues in the CPs and divergent residues that confer specificity for the A03, RA2.7, and M134 TCRs respectively. Affinity measurements confirm the structural data.

We have clarified this point in the revision (Figure 5d and 5e, page 9)

- Label x-axis in Figure 5d: mutations on A03 TCR
- Label x-axis in Figure 5e (top panel): mutations on HLA-DR4, (bottom panel): footprint of A03 TCR docking on HLA-DR4 presenting vimentin-64Cit.
- We also clarified the text (Page 9): “These four residues were involved in pHLA interaction as well as inter-CDR α contacts, as shown in Figure 5b and 5c.”
- Revised text (page 9): “As such, the point mutagenesis study of germline encoded residues have corresponded with our current and previously defined structural data, as shown in Vim-64cit₅₉₋₇₁ restricted A03 TCR and Fib β -74cit₆₉₋₈₁ restricted M134 TCR (Figure 5b and 5c), respectively, highlighting the role of these conserved TRAV6⁺ residues in maintaining the TCR inter-CDR α loop contacts and peptide-HLA-DR4 recognition.”
- We also revised the text (Page 9): “Among the conserved contact residues Glu⁵⁵ α , Gln⁵⁷ α , Gly⁵⁸ α , and Thr⁷⁷ β of HLA-DR4 with in both the A03 and M134 TCRs in pHLA ternary complexes (Figure 5b and 5c), alanine mutation to Gln⁵⁷ α and Thr⁷⁷ β showed ~10-fold reduced affinity compared to wildtype HLA-DR4 recognition while Glu⁵⁵ α showed ~4-fold reduced affinity after mutation (Fig. 5e, Supplementary Fig. 4).”

Reviewer #3:

In the manuscript “The molecular basis underlying T cell specificity towards citrullinated peptides presented by HLA-DR4”, Loh et al identify and molecularly characterize multiple TCRs that recognize a series of citrullinated peptides displayed by HLA-DR4. As citrullinated peptides are a primary driver of disease in rheumatoid arthritis, understanding how these epitopes are recognized is relevant for human health. The work here is well done, with a few outstanding questions about bigger-picture data interpretation.

Comment: A significant portion of this work studies HLA-DR4-restricted TCRs derived from a transgenic mouse. These TCRs largely behave the same as any other TCR, but it is somewhat unclear how much certain features of these TCRs recognizing antigen can directly translate to human TCRs. For example, the observed dominant use of mouse TRAV6 is interesting, but this does not necessarily correspond to a similar mode of recognition or dominant gene usage in human disease. Some discussion about what can – and cannot – be learned in such a system would be helpful for the reader.

We thank reviewer for raising this point. In a previous study on the fibrinogen epitope (Lim et al., *Sci Immunol.* 2021), we discovered a conserved motif of L(DW)G in the mouse TCR repertoire that played critical role in citrullinated fibrinogen epitope recognition. This L(DW)G motif was also present in human TCRs towards the same fibrinogen epitope, and the key contact points between mouse TCR-HLA-DR4-fibcit were conserved in the human TCR- HLA-DR4-fibcit complex, thereby underscoring the relevance of the mouse transgenic work to that of human health. In this study, we studied two different citrullinated epitopes, vimentin and α -enolase. We discovered there is similar mode of recognition between CDR1 α residues of mouse TRAV6⁺ and human TRAV26-1 TCRs with pHLA, thereby further underscoring the relevance of the mouse transgenic to that of human RA.

We provide the following statement in the discussion page 14:

“Intriguingly, the CDR1 α residues (asparagine and tyrosine) either at positions 36 and 38 of TRAV6⁺ TCRs, or vice versa in TRAV26-1 TCR, made similar contacts with P2 of peptide and HLA-DR4 β -chain, thus underscoring the relevance of the mouse transgenic model to that of human RA.”

Comment: While there is apparent similarity between the A03 and A07 TCR structures and therefore shared features in the TRAV6 gene usage, I have difficulty seeing how the similarity extends to the M134 TCR recognizing the fibrinogen-derived epitope – the contacts look quite different, and indeed the authors state the alpha chain as a whole contributes less to the M134 interface. If TRAV6 is regularly used in these other epitopes, that is interesting – but the ‘why’ of it could be more clearly articulated.

To clarify, while the contacts are indeed different, analogous to the A03 and A07 TCRs, the TRAV6⁺ M134 TCR also plays a role in peptide-HLA-DR4 recognition as well as maintaining the TCR inter-CDR α loop contacts (Figure 5b and 5c). The TRAV-peptide recognition is dominated at position 2 of peptide, which is one of the key residues involved in TCR repertoire selection (Figure 7e, stated in page12 in the manuscript).

We now revise the text (Page 9):

“As such, the point mutagenesis study of germline encoded residues have corresponded with our current and previously defined structural data, as shown in Vim-64^{cit59-71} restricted A03 TCR and Fib β -74^{cit69-81} restricted M134 TCR (Figure 5b and 5c), respectively, highlighting the role of these conserved TRAV6⁺ residues in maintaining the TCR inter-CDR α loop contacts and peptide-HLA-DR4 recognition.”

Comment: Moving to the human TCR data, the amino acid differences at P2 along the peptides is interesting, but it should be noted that each of the other TCR contacts differ – it therefore seems overstated to ascribe the specificity differences so strongly to the P2 differences.

To clarify, it was shown in this study and a previous publication that the TCR clones are very specific to the restricted epitope presented by HLA-DR4 (Figure 1d and Lim et al., Sci Immunol. 2021). Consistent with the tetramer staining data, our structural findings in vimentin-, alpha-enolase-, and previously solved fibrinogen-restricted pHLA complexes showed that the solvent exposed residues were at positions P2, P5, P7 and P8 of the bound peptide. Residues in P5 and P8 consist of relatively small side chain with no significant charge feature or bulky side chain. The P7 residue on the other hand is pointed towards HLA-DR4, as shown in figure 7h, making little or no contact with TCR. The most salient observation relates to the distinct chemical features of P2 in all three epitopes, in which the negatively charge aspartate (α -enolase), long protruding positively charged side chain of arginine (fibrinogen), and small non-polar alanine (vimentin) that pointed towards TCR and thus, we contend, define the specificity difference.

We now state in the discussion (page 15)

“Positions P2, P5, P7 and P8 of the citrullinated epitopes are solvent exposed in their respective pHLA complexes and thus represent potential TCR contact points, However, it was the charge characteristic and side chain features of the P2 residues of the peptide that appeared to play an important role in determining the T cell repertoire selected for a given citrullinated neo-antigen,”

Comment: Minor point – it would be good to standardize the boundaries and format for showing CDR loops in Figures 1, 2, and 5.

We have now revised the format and CDR sequence boundaries of Figure 2c and 5a.

REVIEWERS' COMMENTS

Reviewer #1 (Remarks to the Author):

As revised, the manuscript is acceptable for publication. All of my stated concerns have been adequately allayed.

Reviewer #2 (Remarks to the Author):

The minor issues that I had have been thoroughly addressed in this revision.

Reviewer #3 (Remarks to the Author):

My comments have been sufficiently addressed, thank you